# IMSE: Intrinsic Mixture of Spectral Experts Fine-tuning for Test-Time Adaptation

**Sunghyun Baek**[1]    **Jaemyung Yu**[1]    **Seunghee Koh**[1]
**Minsu Kim**[2]    **Hyeonseong Jeon**[2]    **Junmo Kim**[1*]

[1]Korea Advanced Institute of Science and Technology (KAIST)
[2]LG Energy Solution
`{baeksh, jaemyung, seunghee1215}@kaist.ac.kr`
`{mskim0410, cutz}@lgensol.com, junmo.kim@kaist.ac.kr`

## Abstract

Test-time adaptation (TTA) has been widely explored to prevent performance degradation when test data differ from the training distribution. However, fully leveraging the rich representations of large pretrained models with minimal parameter updates remains underexplored. In this paper, we propose Intrinsic Mixture of Spectral Experts (IMSE) that leverages the spectral experts inherently embedded in Vision Transformers. We decompose each linear layer via singular value decomposition (SVD) and adapt only the singular values, while keeping the singular vectors fixed. We further identify a key limitation of entropy minimization in TTA: it often induces feature-collapse, causing the model to rely on domain-specific features rather than class-discriminative features. To address this, we propose a diversity maximization loss based on expert–input alignment, which encourages diverse utilization of spectral experts during adaptation. In the continual test-time adaptation (CTTA) scenario, beyond preserving pretrained knowledge, it is crucial to retain and reuse knowledge from previously observed domains. We introduce Domain-Aware Spectral Code Retrieval, which estimates input distributions to detect domain shifts, and retrieves adapted singular values for rapid adaptation. Consequently, our method achieves state-of-the-art performance on various distribution-shift benchmarks under the TTA setting. In CTTA and Gradual CTTA, it further improves accuracy by 3.4 percentage point (pp) and 2.4 pp, respectively, while requiring 385 times fewer trainable parameters. Our code is available in `https://github.com/baek85/IMSE`.

## 1 Introduction

Real-world data often deviates from the training distribution, leading to performance degradation in deployed models. Test-time adaptation (TTA) mitigates this distribution shift by adapting a source-pretrained model to unseen target domains in an online manner, without access to the source data. However, TTA is prone to forgetting task-relevant knowledge as adaptation progresses under continuously shifting data distributions. To address this more realistic scenario, TTA has been extended to continual test-time adaptation (CTTA). Existing approaches differ mainly in the extent of parameter updates. Some methods restrict adaptation to normalization statistics or affine parameters (Ioffe & Szegedy, 2015; Wang et al., 2020; Niu et al., 2022), ensuring stability and efficiency but limiting adaptation capacity. Others allow broader parameter updates or introduce lightweight architectural components such as prompts or adapters (Wang et al., 2022; Yu et al., 2024; Tang et al., 2024; Liu et al., 2024), improving flexibility at the risk of error accumulation, catastrophic forgetting, or additional inference overhead.

We identify three key limitations in existing TTA and CTTA methods. First, how to fully exploit the rich representational capacity inherent in large pre-trained models remains underexplored. Second, in label-free TTA scenarios, entropy minimization often leads the model to exploit domain-specific

---

*Correspondence to Junmo Kim (junmo.kim@kaist.ac.kr).

features rather than class-discriminative features, thereby degrading performance. Third, in CTTA settings, it is crucial not only to preserve pretrained knowledge but also to retain information from previously encountered domains. However, efficient methods for preserving such domain-specific knowledge is not yet well addressed.

To address those limitations, we propose **Intrinsic Mixture of Spectral Experts (IMSE)** with three components: (1) *Intrinsic mixture of spectral experts*: we decompose each linear layer via SVD and interpret the resulting rank-1 components as spectral experts with distinct functional roles. From this perspective, a linear layer can be reinterpreted as an intrinsic mixture of spectral experts, where the singular values correspond to the contribution of each expert. We adapt only the singular values while keeping the orthogonal bases fixed, to utilize the pretrained feature extractors. (2) *Diversity maximization loss*: we maximize diversity of response patterns of spectral experts, ensuring that the model does not over-extract domain-specific features during adaptation, even in the absence of labels. (3) *Domain-aware spectral code retrieval*: we explicitly preserve and reuse domain knowledge by storing adapted singular values together with domain descriptors in a domain bank, thereby mitigating domain knowledge forgetting in CTTA.

We demonstrate IMSE under TTA , CTTA, and Gradual CTTA setting. Experiments with MAE (He et al., 2022) and CLIP (Radford et al., 2021) further show its generalization ability in the TTA setting. Furthermore, it achieves superior performance over CTTA baselines with significantly fewer trainable parameters and faster runtime.

Our main contributions are summarized as follows:

- We propose IMSE, a TTA framework that reinterprets linear layers of pretrained models as intrinsic mixture of spectral experts. By fine-tuning only singular values, IMSE enables parameter-efficient adaptation while preserving the pretrained feature extractors.
- We introduce a diversity maximization loss that compensates for feature collapse caused by entropy minimization, ensuring that pretrained feature extractors are effectively utilized even in the absence of labels.
- We design a domain-aware spectral code retrieval mechanism to mitigate domain knowledge forgetting, enabling rapid adaptation in continual test-time adaptation.
- We proivde state-of-the-art performance in TTA, CTTA, and Gradual CTTA settings and is further validated on diverse pretrained models, including MAE and CLIP, under the TTA setting.

## 2 RELATED WORK

**Test-time adaptation.** To preserve the class-discriminative capability of pretrained models during adaptation in classification tasks, various methods optimize different sets of parameters and consequently adopt different objective functions. Early approaches based on entropy minimization avoid data augmentation and use only the predictions on the current test data to define the loss. TENT (Wang et al., 2020) fine-tunes only the modulation parameters of the Batch Normalization (Ioffe & Szegedy, 2015) layer to minimize prediction entropy. Building on this idea, EATA (Niu et al., 2022) improves stability by filtering out unreliable high-entropy samples, while SAR (Niu et al., 2023) incorporates sharpness-aware optimization for more stable adaptation. Other works introduce pseudo-label refinement techniques that leverage the average of predictions obtained from multiple data augmentations and use an Exponential Moving Average (EMA) updated teacher model to further stabilize training. This strategy is typically employed in approaches closer to full parameter fine-tuning, such as CoTTA (Wang et al., 2022) and ViDA (Liu et al., 2024). Other lines of research (Tang et al., 2024; Lee et al., 2024) keep most of the backbone intact while introducing architectural modifications, for example by inserting lightweight modules such as adapters or domain-specific tokens.

**Singular Value Fine-tuning.** Recent studies exploit singular-value fine-tuning for parameter-efficient fine-tuning of large models, such as SVFT (Lingam et al., 2024) and SVDiff (Han et al., 2023), which update only the singular values of decomposed layers of a Large Language Model (LLM) and a Diffusion Model, respectively. Other works, including PiSSA (Meng et al., 2024) and MiLoRA (Wang et al., 2025), leverage singular value decomposition (SVD) to improve low-rank Adaptation (LoRA) (Hu et al., 2022) initialization for LLM fine-tuning.

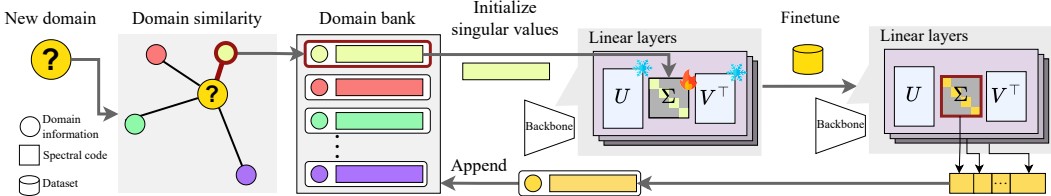

Figure 1: **IMSE-Retrieval with domain bank.** When adapting to a new domain, we select initial singular values based on domain similarity using the domain descriptor, then fine-tune the $\boldsymbol{\sigma}$ components within linear layers. The adapted spectral code $\boldsymbol{S}$ is stored in the Domain Bank, and this process repeats for subsequent domains. Note that domain descriptors are designed to estimate the distribution of test data.

Our method not only fine-tunes the singular values of linear layers for test-time adaptation, but also maximizes feature diversity using singular vectors, and introduces a retrieval mechanism that reuses the adapted singular values for continual test-time adaptation.

## 3 INTRINSIC MIXTURE OF SPECTRAL EXPERTS

We propose Intrinsic Mixture of Spectral Experts (IMSE) that adapts pretrained models to the current domain while preserving the independent base extractors intrinsically obtained during pretraining. To compensate for the reduction of feature diversity caused by entropy minimization, we introduce a diversity maximization loss that encourages diverse utilization of spectral experts. In addition, we introduce Domain-Aware Spectral Code Retrieval, a mechanism that preserves and reuses domain-specific adapted singular values. This retrieval process enables a stable and faster adaptation process using the most relevant spectral code. The overall framework of IMSE-Retrieval, which includes the retrieval process, is illustrated in Figure 1.

### 3.1 INTRINSIC MIXTURE OF SPECTRAL EXPERTS

**Spectral experts and spectral code.** We adapt the model by applying SVD to each linear layer, thereby adjusting the degree of contribution of diverse pretrained feature extractors to the final output. Given a linear transformation $\mathbf{W}^{(l)}$ with output dim of $d_{out}^{(l)}$ and input dim of $d_{in}^{(l)}$ in the $l$-th layer, SVD factorizes a $r^{(l)}$-rank matrix $\mathbf{W}^{(l)}$ into a left singular vector matrix $\mathbf{U}^{(l)} \in \mathbb{R}^{d_{out}^{(l)} \times r^{(l)}}$, a diagonal matrix of singular values $\boldsymbol{\Sigma}^{(l)} \in \mathbb{R}^{r^{(l)} \times r^{(l)}}$, and a right singular vector matrix $\mathbf{V}^{(l)} \in \mathbb{R}^{d_{in}^{(l)} \times r^{(l)}}$. This decomposition is formally expressed as:

$$\mathbf{W}^{(l)} = \mathbf{U}^{(l)} \boldsymbol{\Sigma} \mathbf{V}^{(l)\top} = \sum_{i=1}^{r^{(l)}} \sigma_i^{(l)} \mathbf{u}_i^{(l)} \mathbf{v}_i^{(l)\top}, \tag{1}$$

where each rank-1 component is defined by orthonormal basis vectors $\mathbf{u}_i^{(l)}$ and $\mathbf{v}_i^{(l)}$, and scaled by the singular value $\sigma_i^{(l)}$, which is non-negative and sorted in non-increasing order.

We refer to the subset of these rank-1 components $\mathbf{u}_i \mathbf{v}_i^\top$ as an $i$-th *spectral expert* and define the *spectral code* as the set of singular values across all $L$ layers, represented as follows:

$$\boldsymbol{S} = \{\boldsymbol{\sigma}^{(l)}\}_{l=1}^L, \quad \boldsymbol{\sigma}^{(l)} = [\sigma_1^{(l)}, \dots, \sigma_{r^{(l)}}^{(l)}] \tag{2}$$

An input vector $\mathbf{x}^{(l)}$ is projected by each spectral expert through orthogonal bases $\mathbf{u}_i^{(l)}$ and $\mathbf{v}_i^{(l)}$. Since the singular vectors are mutually orthogonal, the outputs from different experts are also orthogonal, i.e., $(\mathbf{u}_i^{(l)} \mathbf{v}_i^{(l)\top} \mathbf{x}^{(l)})^\top (\mathbf{u}_j^{(l)} \mathbf{v}_j^{(l)\top} \mathbf{x}^{(l)}) = 0$ for $i \neq j$.

We regard the linear layer as a mixture of $r^{(l)}$ spectral experts that generate orthogonal outputs given the same input. Based on the interpretation that singular values determine each spectral expert's contribution weight, we adapt the model to new domains by updating only the singular values while freezing the singular vectors. Fine-tuning singular values not only allows preserving the subspace of

$\mathbf{W}^{(l)}$, but also leverages the rich feature extractors that pretrained models have already acquired from diverse training data.

**Expert-Input alignment statistics.**    The output direction of each spectral expert is determined by the fixed left singular vector $\mathbf{u}^{(l)}_{(i)}$, while its response strength to an input $\mathbf{x}^{(l)}_n$ is given by the scalar $\sigma^{(l)}_{(i)} \mathbf{v}^{(l)\top}_{(i)} \mathbf{x}^{(l)}_n$. In TTA, entropy minimization induces feature collapse, where only a subset of experts is dominantly activated, causing the model to focus on domain-specific rather than class-discriminative features. To quantify this collapse, we introduce expert-input alignment statistics measuring how strongly each expert responds to the test data.

Given $B$ samples with $T$ tokens in a batch, yielding $N = B \times T$ input vectors, let the alignment of $n$-th input vector $\mathbf{x}^{(l)}_n \in \mathbb{R}^{d^{(l)}_{in}}$ with each right singular vector $\mathbf{v}^{(l)}_i$ as $\mathbf{a}^{(l)}_{n,i} = \frac{\mathbf{v}^{(l)\top}_i \mathbf{x}^{(l)}_n}{\|\mathbf{x}^{(l)}_n\|_2}$, which isolates the directional alignment from the magnitude. Based on this, we define two alignment-based statistics per expert:

$$\bar{\phi}^{(l)}_i = \frac{1}{N} \sum_{n=1}^{N} \mathbf{a}^{(l)}_{n,i}, \quad \mathrm{Std}^{(l)2}_i = \frac{1}{N} \sum_{n=1}^{N} \left( \mathbf{a}^{(l)}_{n,i} - \bar{\phi}^{(l)}_i \right)^2. \tag{3}$$

Here, $\bar{\phi}^{(l)}_i$ measures the mean of the $i$-th expert across input tokens, while $\mathrm{Std}^{(l)}_i$ quantifies the utilization diversity, indicating how differently the expert responds to different tokens. Low $\mathrm{Std}^{(l)}_i$ suggests that the expert captures domain-specific patterns shared across tokens within the batch.

**Diversity Maximization Loss.**    We propose a diversity maximization loss that encourages the model to generate diverse features by using expert-input alignment statistics

$$\mathcal{L}_{dm} = - \sum_{l \in \Lambda_{dm}} \frac{1}{r^{(l)}} \sum_{i=1}^{r^{(l)}} \mathrm{Std}^{(l)}_i, \tag{4}$$

where $\Lambda_{dm}$ is a set of selected layers, and $r^{(l)}$ is the number of experts in layer $l$. We choose the latter layers as the elements of $\Lambda_{dm}$ since the diversity pattern tends to decrease in layers closer to the classification head. We provide related anlaysis in Appendix D.

**Optimization.**    We optimize the spectral code $S$ using an entropy minimization objective $\mathcal{L}_{entmin}$ incorporating entropy-based sample filtering as in SAR (Niu et al., 2023).

The objective aims to minimize the prediction entropy $H(\hat{y}) = - \sum_c p(\hat{y}_c) \log p(\hat{y}_c)$, where $\mathbf{x}$ is an input sample of the test domain, $\hat{y} = f(\mathbf{x})$ is the model prediction, and $p(\hat{y}_c)$ denotes the predicted probability of class $c$. The full entropy minimization loss is defined as $\mathcal{L}_{entmin}(\mathbf{x}) = \mathbb{I}_{\{\mathbf{x} \in S(\mathbf{x})\}} H(\hat{y})$, where $S(\mathbf{x})$ is the sample selection function that identifies reliable samples, such as low-entropy samples. The indicator function $\mathbb{I}(\cdot)$ masks out uncertain samples.

We jointly use the entropy minimization and diversity maximization objectives as follows:

$$\mathcal{L}_{IMSE} = \mathcal{L}_{entmin} + \lambda_{dm} \cdot \mathcal{L}_{dm}. \tag{5}$$

We also adopt Sharpness-Aware Minimization (Foret et al., 2021) to enhance stability, following SAR (Niu et al., 2023). Further details are provided in Appendix A.

## 3.2    DOMAIN-AWARE SPECTRAL CODE RETRIEVAL

Fine-tuning only the singular values yields a compact spectral code that succinctly captures domain-specific knowledge. We exploit compactness of the spectral code in **IMSE-Retrieval**, which retrieves stored spectral codes from past domains to enable fast adaptation to new test-time domains in CTTA. To represent the input distribution, we employ a lightweight *domain descriptor*, computed by extracting patch tokens after the patch and position embedding stage and computing their channel-wise mean and variance $\phi = \{mean, variance\}$ across batch and token dimension. At the $t$-th adaptation step, the descriptor $\phi'_t$ for each input is accumulated as $\phi_{(t)}$, using an exponential moving average (EMA), where $\phi_{(t)} = \alpha \phi_{(t-1)} + (1 - \alpha)\phi'_t$. In the following, we detail the initialization and update of the domain bank, and the retrieval process during CTTA.

**Domain Bank.** The *domain bank* is a memory module that stores pairs of domain descriptors and their corresponding spectral codes, i.e., $[\phi^k, \boldsymbol{S}^k]$ for each encountered domain $k$. It serves as a repository of domain-specific knowledge from previously encountered domains during CTTA that can be retrieved to initialize model adaptation whenever a new domain is detected. Prior to continual test-time adaptation, we initialize the domain bank using information from the source domain. The original singular values $\boldsymbol{S}_{\text{pre}}$ are stored in the domain bank together with the aggregated domain descriptors $\phi$ as $[\phi_1, \boldsymbol{S}_1]$. This source entry serves as the initial anchor point for adaptation. At the beginning of continual test-time adaptation, we initialize $\boldsymbol{S}$ using $\boldsymbol{S}_{\text{pre}}$. Once the first domain shift is detected during adaptation, we store the adapted spectral code and the updated EMA descriptor as the second entry in the domain bank as $[\phi_2, \boldsymbol{S}_2]$.

**Spectral Code Retrieval.** We detect new domains and reuse the adapted singular values guided by the current domain descriptor. To identify a domain shift, we compare the current input-level descriptor $\phi$ with the accumulated descriptor $\phi_{(t)}$ using a symmetric Kullback–Leibler (KL) divergence (Kullback & Leibler, 1951)across $C$ channels:

$$D(\phi_1, \phi_2) = \tfrac{1}{C} \sum_{i=1}^{C} \big[ KL(\phi_{1,i} \, \| \, \phi_{2,i}) + KL(\phi_{2,i} \, \| \, \phi_{1,i}) \big]. \tag{6}$$

A domain shift is detected when $D(\phi'_t, \phi_{(t)})$ exceeds a predefined threshold $\tau$. The stabilized descriptor $\phi_{(t)}$ and its corresponding adapted spectral code $\boldsymbol{S}_{(t)}$ are then stored in the domain bank to represent the corresponding domain. We retrieve the most similar previously encountered domain by comparing the current descriptor $\phi'_t$ with the stored set $\{\phi_k\}_{k=1}^{K}$, where $K$ is the number of recorded domains, and select $k^* = \arg\min_k D(\phi'_t, \phi_k)$. The singular values associated with the matched domain $k^*$ initialize adaptation for the current input, and are further refined during subsequent updates. This retrieval and initialization process is triggered automatically whenever a new domain is detected.

## 4 EXPERIMENTS

### 4.1 SETUP

**Datasets.** We evaluate our method on ImageNet-C (Hendrycks & Dietterich, 2019), which includes 15 corruption types categorized into Noise, Blur, Weather, and Digital. Additionally, we use ImageNet-R (Hendrycks et al., 2021a) (renditions) and ImageNet-A (Hendrycks et al., 2021b) (adversarial examples) to test robustness against different distribution shift types. Implementation details are provided in the Appendix A.

**Single domain test-time adaptation.** We use the full validation set (50,000 images) from ImageNet-C, where each corruption type is evaluated independently. Unless otherwise specified, we use a standard ViT model (Dosovitskiy et al., 2021) fine-tuned on ImageNet-1k (Krizhevsky et al., 2012) as the default backbone. To assess the generalizability of our method under different pretraining strategies, we use Vision Transformers pretrained via MAE (He et al., 2022) and CLIP (Radford et al., 2021). For the CLIP-pretrained variant, we utilize the fine-tuning model from (Cherti et al., 2023), which is finetuned on ImageNet-12k and subsequently on ImageNet-1k. Beyond standard corruptions, we assess adaptation performance under different distribution shift types using ImageNet-R and ImageNet-A, which contain artistic rendering images and naturally occurring adversarial samples, respectively. These datasets present fundamentally different challenges compared to ImageNet-C.

**Continual test-time adaptation.** Following ViDA (Liu et al., 2024), we use 5,000 images from the ImageNet-C validation set provided by RobustBench (Croce et al., 2021). The model continuously adapts to the incoming test data stream without access to domain boundary information.

**Gradual continual test-time adaptation.** To assess whether IMSE can detect and adapt to subtle distribution changes, we construct a gradual-shift evaluation scenario using ImageNet-C. Following the corruption order used in our main CTTA experiments (e.g., Gaussian Noise $\rightarrow$ Shot Noise $\rightarrow$ Impulse Noise $\rightarrow$ Defocus Blur $\rightarrow$ $\cdots$ $\rightarrow$ JPEG Compression), we generate a continuous sequence in which the severity level of each corruption smoothly varies as: $1 \rightarrow 2 \rightarrow 3 \rightarrow 4 \rightarrow 5 \rightarrow 4 \rightarrow 3 \rightarrow 2 \rightarrow 1$. Repeating this process for all 15 corruption types results in 135 sequential test domains.

Table 1: **Test-time adaptation on ImageNet-C (50k)**. Accuracy(%)(↑) over 15 corruptions at severity level 5 using ViT-Base models with different pretraining strategies. Results with * are reproduced by us.

| Pretrain | Method | Noise | | | Blur | | | | Weather | | | | Digital | | | | Avg.(↑) |
|---|---|---|---|---|---|---|---|---|---|---|---|---|---|---|---|---|---|
| | | Gauss. | Shot | Impul. | Defoc. | Glass | Motion | Zoom | Snow | Frost | Fog | Brit. | Contr. | Elastic | Pixel. | JPEG | |
| Supervised | Source | 46.9 | 47.6 | 46.9 | 42.7 | 34.2 | 50.5 | 44.7 | 56.9 | 52.6 | 56.5 | 76.1 | 31.8 | 46.7 | 65.5 | 66.0 | 51.0 |
| | T3A | 16.6 | 11.8 | 16.4 | 29.9 | 24.3 | 34.5 | 28.5 | 15.9 | 27.0 | 49.1 | 56.1 | 44.8 | 33.3 | 45.1 | 49.4 | 32.2 |
| | CoTTA | 40.3 | 31.8 | 39.6 | 35.5 | 33.1 | 46.9 | 37.3 | 2.9 | 46.4 | 59.1 | 71.7 | 55.5 | 46.4 | 59.4 | 59.0 | 44.4 |
| | DDA | 52.5 | 54.0 | 52.1 | 33.8 | 40.6 | 33.3 | 30.2 | 29.7 | 35.0 | 5.0 | 48.6 | 2.7 | 50.0 | 60.0 | 58.8 | 39.1 |
| | MEMO | 58.1 | 59.1 | 58.5 | 51.6 | 41.2 | 57.1 | 52.4 | 64.1 | 59.0 | 62.7 | **80.3** | 44.6 | 52.8 | 72.2 | 72.1 | 59.1 |
| | AdaContrast | 54.4 | 55.8 | 55.8 | 52.5 | 42.2 | 58.7 | 54.3 | 64.6 | 60.1 | 66.4 | 76.8 | 53.7 | 61.7 | 71.9 | 69.6 | 59.9 |
| | CFA | 56.9 | 58.0 | 58.1 | 54.4 | 48.9 | 59.9 | 56.6 | 66.4 | 64.1 | 67.7 | 79.0 | 58.8 | 64.3 | 71.7 | 70.2 | 62.4 |
| | TENT | 57.6 | 58.9 | 58.9 | 57.6 | 54.3 | 61.0 | 57.5 | 65.7 | 54.1 | 69.1 | 78.7 | 62.4 | 62.5 | 72.5 | 70.6 | 62.8 |
| | DePT-G | 53.7 | 55.7 | 55.8 | 58.2 | 56.0 | 61.8 | 57.1 | 69.2 | 66.6 | 72.2 | 76.3 | 63.2 | 67.9 | 71.8 | 68.2 | 63.6 |
| | SAR | 58.0 | 59.2 | 59.0 | 58.0 | 54.7 | 61.2 | 57.9 | 66.1 | 64.4 | 68.6 | 78.7 | 62.4 | 62.9 | 72.5 | 70.5 | 63.6 |
| | EATA | 54.8 | 55.3 | 55.6 | 58.0 | 59.1 | 63.4 | 61.5 | 67.7 | 66.2 | _73.2_ | 77.9 | **68.0** | 68.4 | 73.1 | 70.3 | 64.8 |
| | DPAL | _59.8_ | _61.7_ | 61.0 | _59.1_ | 60.5 | 64.9 | 63.8 | _70.2_ | _68.9_ | 72.6 | 79.7 | 62.6 | _70.9_ | _75.6_ | _73.1_ | _67.0_ |
| | IMSE | **61.9** | **63.9** | **63.4** | **61.8** | **62.5** | **67.5** | **68.1** | **71.9** | **70.2** | **74.4** | _79.8_ | _67.6_ | **72.9** | **76.2** | **73.5** | **69.0** |
| MAE | Source | 56.8 | 56.8 | 57.5 | 46.9 | 35.6 | 53.1 | 44.8 | 62.2 | 62.6 | 65.7 | 77.7 | 32.6 | 46.0 | 67.0 | 67.6 | 55.5 |
| | TENT* | 60.2 | 61.4 | 61.8 | 59.1 | 56.7 | 63.5 | 59.0 | 59.0 | 64.1 | 3.3 | _79.2_ | _67.6_ | 60.9 | 72.9 | 70.6 | 60.0 |
| | SAR* | 59.2 | 60.3 | 60.7 | 57.4 | 55.8 | 61.7 | 57.6 | _66.0_ | 63.7 | 66.1 | 78.7 | 64.5 | 62.3 | 72.4 | 70.1 | 63.8 |
| | DPAL* | _61.6_ | _63.2_ | **63.4** | _59.4_ | _59.7_ | _64.6_ | _61.5_ | 63.0 | **70.4** | 63.7 | 79.0 | 60.9 | _69.1_ | _74.9_ | _72.6_ | _65.9_ |
| | IMSE | **61.9** | **63.8** | **63.4** | **61.3** | **61.2** | **66.7** | **65.6** | **71.2** | _68.4_ | **72.9** | **80.2** | **68.4** | _70.8_ | **75.7** | **73.4** | **68.3** |
| CLIP | Source | 35.2 | 36.6 | 38.7 | 31.8 | 25.6 | 43.9 | 36.2 | 51.6 | 50.3 | 46.9 | 76.6 | 21.6 | 34.3 | 51.7 | 60.8 | 42.8 |
| | TENT* | _55.4_ | _57.6_ | **57.6** | 50.7 | 51.9 | 59.5 | 54.2 | 64.6 | 57.5 | 68.5 | **80.1** | _63.0_ | 6.8 | 72.3 | 70.0 | 58.0 |
| | SAR* | 54.9 | 57.0 | 57.1 | 50.5 | 53.4 | 59.7 | _55.3_ | 64.5 | 60.6 | 67.7 | 78.2 | 62.3 | 63.1 | 71.6 | 69.4 | 61.7 |
| | DPAL* | 55.0 | 57.3 | 57.0 | _51.4_ | _54.0_ | 60.0 | 50.9 | _65.9_ | _62.7_ | _69.2_ | 79.7 | 62.8 | _64.8_ | _72.8_ | 70.7 | _62.3_ |
| | IMSE | **55.5** | **57.7** | **57.6** | **55.3** | **58.2** | **65.5** | **65.4** | **69.1** | **65.7** | **71.8** | _79.9_ | 64.6 | **70.3** | **74.1** | **71.4** | **65.5** |

## 4.2 BASELINES

**Single Domain TTA.** We adopt the baselines used in DPAL (Tang et al., 2024), including T3A (Iwasawa & Matsuo, 2021), CoTTA (Wang et al., 2022), DDA (Gao et al., 2023), MEMO (Zhang et al., 2022), AdaContrast (Chen et al., 2022), CFA (Kojima et al., 2022), TENT (Wang et al., 2020), DePT-G (Gao et al., 2022), SAR (Niu et al., 2023), and EATA (Niu et al., 2022).

**Continual TTA and Gradual Continual TTA.** We primarily compare with TENT (Wang et al., 2020), CoTTA (Wang et al., 2022), and ViDA (Liu et al., 2024), the most widely adopted baselines with public implementations for continual TTA on ImageNet-C. For fair comparison, we adopt standard ViT normalization (mean = [0.5, 0.5, 0.5], std = [0.5, 0.5, 0.5]) in our main experiments, noting that some methods like ViDA don't use normalization in their official implementations.

## 4.3 EXPERIMENTS ON VARIOUS TTA SETTINGS

We analyze the effectiveness of IMSE through single-domain test-time adaptation experiments on the ImageNet-C dataset. In Table 1, our method shows state-of-the-art performance across all three pretraining strategies. The performance gap becomes more emphasized with MAE and CLIP pretraining. IMSE outperforms DPAL by 3.4 percentage points (pp) and 2.8 pp, respectively. These experimental results indicate that adjusting only the singular values is effective regardless of the pretraining method.

We further evaluate the robustness of our method under different types of distribution shift by using ImageNet-R and A. In Table 2, our method demonstrates strong adaptation performance on both datasets. Specifically, our method outperforms DPAL (Tang et al., 2024) by 5.0 pp on ImageNet-R and by 4.9 pp on ImageNet-A. Additionally, effectiveness of IMSE under more challenging domain shifts (ImageNet-3DCC, OfficeHome, and DomainNet) is reported in Appendix G.

## 4.4 CONTINUAL TEST-TIME ADAPTATION

We compare the performance of the IMSE-Retrieval and baselines in the continuously changing CTTA scenario. As shown in Table 4, IMSE-Retrieval shows the best performance. Our method consistently outperforms other baselines across every corruption domain, yielding an average improvement of 6.7 percentage points (pp) over ViDA. In particular, during the transition from noise to blur corruptions, IMSE-Retrieval surpasses ViDA by 7.2 pp on Defocus Blur and by a substantial margin of 15.2 pp on Glass Blur. Furthermore, except for the Brightness corruption, our method outperforms ViDA by a

Table 2: **Test-time adaptation on ImageNet-R/A**. Accuracy (%)(↑) on two different datasets using supervised pretrained ViT-Base model.

| Method | ImageNet-R | ImageNet-A |
|--------|------------|------------|
| Source | 57.2 | 31.1 |
| TENT | 61.3 | 44.5 |
| SAR | 62.0 | 45.3 |
| DPAL | 64.8 | 49.9 |
| IMSE | **69.8** | **54.8** |

Table 3: **Gradual CTTA on ImageNet-C.** Results with * are reproduced by us.

| Method | Avg Acc. (%)(↑) |
|--------|-----------------|
| Source | 67.3 |
| TENT[*] | 70.7 |
| CoTTA[*] | 69.5 |
| ViDA[*] | 72.5 |
| IMSE-Retrieval | **74.9** |

Table 4: **Continual test-time adaptation on ImageNet-C (5k)**. Accuracy(%)(↑) over 15 corruptions at severity level 5 using a supervised ViT-Base model. Results with * are reproduced by us.

| | Noise | | | Blur | | | | Weather | | | | Digital | | | | |
|--|--|--|--|--|--|--|--|--|--|--|--|--|--|--|--|--|
| ImageNet-C (5k) | Gauss. | Shot | Impul. | Defoc. | Glass | Motion | Zoom | Snow | Frost | Fog | Brit. | Contr. | Elastic | Pixel. | JPEG | Avg.(↑) |
| Source | 46.8 | 46.7 | 46.7 | 44.1 | 31.1 | 48.6 | 44.5 | 54.3 | 50.4 | 54.9 | 75.3 | 31.3 | 45.2 | 65.8 | 66.1 | 50.1 |
| TENT[*] | 48.0 | 50.2 | 51.2 | 45.5 | 34.0 | 51.4 | 46.9 | 57.3 | 54.6 | 56.9 | 76.3 | 39.5 | 46.4 | 66.9 | 66.7 | 52.8 |
| CoTTA[*] | 47.0 | 47.1 | 47.3 | 44.7 | 31.6 | 49.0 | 45.2 | 54.8 | 51.0 | 55.5 | 75.9 | 32.0 | 45.9 | 66.1 | 66.7 | 50.7 |
| ViDA[*] | 50.5 | 56.3 | 56.5 | 50.8 | 42.6 | 56.5 | 52.2 | 61.6 | 58.3 | 63.0 | 77.1 | 55.2 | 49.8 | 67.1 | 68.0 | 57.7 |
| IMSE Retrieval | **57.3** | **60.4** | **59.7** | **58.0** | **57.8** | **60.7** | **60.8** | **68.2** | **66.3** | **68.5** | **78.2** | **59.9** | **67.1** | **74.1** | **69.6** | **64.4** |

considerable margin across all subsequent domains. Remarkably, these gains are achieved without any pseudo-label refinement via data augmentation or EMA teacher models, which are essential components in CoTTA and ViDA.

## 4.5 GRADUAL CONTINUAL TEST-TIME ADAPTATION

Table 3 reports the average accuracy under the gradual-shift setting. IMSE-Retrieval achieves the highest performance of 74.9%, outperforming CTTA baselines such as TENT, CoTTA, and ViDA. We additionally investigate the influence of the domain-change threshold $\tau$. A margin of one step around each transition point is allowed when measuring detection quality to account for the smooth nature of the shifts. When the threshold is increased to a more tolerant value from 0.02 to 0.04, the F1 score of domain-change detection decreases from 0.77 to 0.28, while the overall adaptation performance increases from 74.9% to 75.6%. This trade-off indicates that highly sensitive domain-change detection is not always advantageous, and that choosing a moderate threshold helps achieve strong performance with a compact Domain Bank under Gradual CTTA setting.

## 5 ANALYSIS

**Effect of Singular Value Selection Strategy.** To further understand the contribution of different spectral components in TTA, we selectively fine-tune subsets of singular values. While our main approach updates all singular values in each linear layer, this experiment investigates the performance impact of tuning only the top-$R\%$ or bottom-$R\%$ singular values, keeping the rest fixed. As shown in Figure 2a, tuning the top 50–90% of singular values achieves comparable or even slightly better performance than updating all singular values. In contrast, tuning only the bottom-$R\%$ consistently leads to notable performance degradation. For example, when $R$ is 80, updating the top 80% yields the best accuracy of 69.1%, while tuning the bottom 80% results in lower performance of 68.4%. As $R$ decreases, the difference between the top and bottom components becomes more pronounced. When only 20% of the singular values are tunable, the top components still maintain reasonable performance at 67.7%, whereas the bottom components drop sharply to 64.7%. These results demonstrate that spectral experts associated with high-magnitude singular values are more informative and play a dominant role in adaptation under distribution shift.

**The Role of Diversity Maximization Loss.** In this section, we examine how feature diversity in the ViT's last-block output correlates with adaptation performance. Figure 2b shows feature diversity, measured as the mean of the dimension-wise standard deviations of token representations within batch samples. Figure 2c shows the corresponding adaptation performance under Gaussian noise corruption at severity level 5.

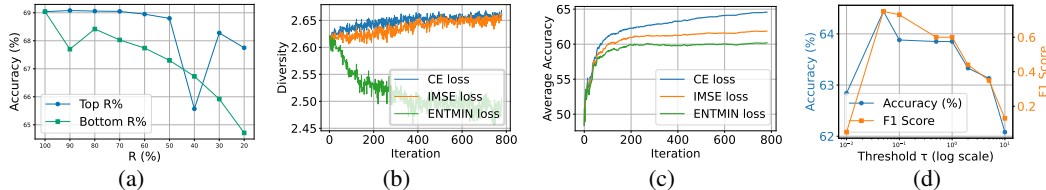

Figure 2: (a) Comparison of top-$R\%$ vs. bottom-$R\%$ singular value selection. (b) Feature diversity across various training methods. (c) Adaptation performance across various training methods. (d) Impact of the threshold $\tau$ on domain shift detection and adaptation performance. CE and TTA losses denote $\mathcal{L}_{ce}$ and $\mathcal{L}_{entmin}$, respectively.

Table 5: **Test-time adaptation on ImageNet-C under non-i.i.d. setting.** Average accuracy (%) across 15 corruption domains for different concentration factors $\alpha$.

| Method | $\alpha$=1.0 (Mild) | $\alpha$=0.5 (Moderate) | $\alpha$=0.1 (Severe) | $\alpha$=0.05 (More severe) |
|---|---|---|---|---|
| IMSE w/o $\mathcal{L}_{dm}$ | 67.79 | 67.78 | 67.64 | 67.33 |
| IMSE | **68.92** | **68.88** | **68.62** | **68.41** |
| Improvement | **+1.13** | **+1.10** | **+0.98** | **+1.08** |

We assess the effect of the diversity-maximization loss combined with entropy minimization, as defined in Equation (5), which encourages diverse utilization of spectral experts in later blocks during adaptation. As shown in Figure 2b, maximizing the proposed objective function effectively prevents the collapse of feature diversity and simultaneously boosts adaptation accuracy to 61.9%, while maintaining a diversity trend close to that of the supervised setting. Further analysis of the diversity of alignment across different transformer block depths is provided in Appendix D.

Specifically, *Supervised* adaptation with cross-entropy loss ($\mathcal{L}_{ce}$) shows a steady increase in diversity and achieves higher adaptation performance, suggesting the model learned class-discriminative features for the current domain. In contrast, the model is adapted in an *unsupervised* manner using entropy minimization ($\mathcal{L}_{entmin}$), the diversity of the last block gradually decreases, and the accuracy drops to 60.2%. This diversity collapse under unsupervised training arises because entropy minimization loss enforces high confidence even for uncertain samples, prompting the model to exploit domain-specific patterns dominant across the incoming test data. As a result, the last-block representation captures domain-related rather than class-discriminative information, yielding sub-optimal performance.

**Robustness of Diversity Maximization in Non-i.i.d. Settings** We further assess whether diversity maximization could unintentionally suppress class-discriminative information under extreme label imbalance. Following DELTA (Zhao et al., 2023), we adopt the Dirichlet non-i.i.d. setting, where the concentration factor $\alpha$ controls the degree of label imbalance within batch. We vary $\alpha$ from 1.0 to 0.05, where smaller $\alpha$ indicates a more severe imbalance. As shown in Table 5, incorporating $\mathcal{L}_{dm}$ consistently improves adaptation accuracy by approximately 1.1 pp across all levels of non-i.i.d. shifts. Notably, even under the most severe class imbalance ($\alpha = 0.05$), using the diversity maximization loss boosts performance from $67.33\%$ to $68.41\%$. This stable improvement confirms that diversity maximization effectively complements entropy minimization, maintaining robust representations even when the input distribution is extremely skewed.

**Domain Shift Detection Quality.** To understand the relationship between domain shift detection quality and adaptation performance, we measure the hyperparameter sensitivity of the domain shift detection threshold $\tau$, which controls the sensitivity of domain shift detection. Figure 2d reports classification accuracy and F1 score for domain shift detection quality at various $\tau$ values. Figure 2d shows that higher F1 score generally leads to higher classification performance. Overall, our method maintains competitive accuracy despite reduced domain-shift detection quality and achieves better performance compared to other methods. To further explain this behavior, we analyze two extremes as follows. When $\tau$ is too low, minor fluctuations trigger frequent shift detections, increasing the number of entries in the domain bank. However, because our method reuses previously stored spectral

Table 6: **Comparison of performance and efficiency** across different CTTA methods. Results with * are reproduced by us.

| Method | Acc (%)(↑) | Params.(↓) | Runtime (sec / batch)(↓) |
|---|---|---|---|
| TENT* | 52.8 | 38.4K | **0.31** |
| CoTTA* | 50.7 | 86.4M | 2.52 |
| ViDA* | 57.7 | 14.2M | 3.49 |
| IMSE-Retrieval | **64.4** | **36.8K** | 0.99 |

Table 7: **Effect of domain descriptor** under Continual test-time adaptation on ImageNet-C.

| Method | Avg Acc. (%)(↑) |
|---|---|
| Source | 50.1 |
| IMSE | 62.2 |
| IMSE-Retrieval (Mean) | 63.1 |
| IMSE-Retrieval (Var) | 63.2 |
| IMSE-Retrieval | **64.4** |

Table 8: **Test-time adaptation ablation study on loss components using ImageNet-C.** ✓ indicates the component is used.

| $\mathcal{L}_{\text{entmin}}$ | $\mathcal{L}_{\text{dm}}$ | Avg. Acc. (%)(↑) |
|---|---|---|
| ✓ | - | 67.8 |
| - | ✓ | 32.7 |
| ✓ | ✓ | **69.1** |

Table 9: **Continual test-time adaptation ablation study on components using ImageNet-C.** ✓ indicates the component is used.

| $\mathcal{L}_{\text{entmin}}$ | $\mathcal{L}_{\text{dm}}$ | Domain Bank | Avg. Acc. (%)(↑) |
|---|---|---|---|
| ✓ | - | - | 59.4 |
| ✓ | ✓ | - | 62.2 |
| ✓ | - | ✓ | 62.8 |
| ✓ | ✓ | ✓ | **64.4** |

codes, these redundant entries cause less performance degradation. When $\tau$ is too high, the change of test domain is undetected, leading to suboptimal adaptation and forgetting of previously encountered domains, resulting in a performance drop. These findings emphasize that accurate domain shift detection is crucial not merely for maximizing classification accuracy, but for maintaining an efficient and meaningful domain bank in CTTA.

**Efficiency of IMSE-Retrieval.** To evaluate efficiency, we measure runtime on a single A6000 GPU with a batch size of 64 and analyze IMSE-Retrieval across three aspects: trainable parameters, runtime, and additional storage, as shown in Table 6. First, IMSE-Retrieval achieves the highest performance (64.4%) while requiring only 36.8K trainable parameters, which corresponds to 0.05% of the parameters updated by CoTTA and about 0.26% relative to ViDA. Second, the measured runtime shows that IMSE-Retrieval is 3.5 times faster than ViDA and 2.5 times faster than CoTTA. This efficiency is primarily related to the low computational cost of SVD. The SVD of all linear layers is performed only once as an offline preprocessing step before adaptation, taking approximately 5.0 seconds. During adaptation, reconstructing weights via $W = U\Sigma V^\top$ adds an overhead of 0.07 seconds per batch, which is marginal relative to the 0.99 seconds required for the full adaptation pass (only about 7% of the total adaptation time). Although IMSE-Retrieval is slower than TENT, our approach achieves much higher accuracy. Third, the storage required for the Domain Bank is approximately 0.33 MB per domain, which is negligible relative to the ViT-Base backbone size (330.23 MB). These results demonstrate that IMSE-Retrieval not only provides strong adaptation performance for CTTA but also maintains high efficiency across all measured aspects.

## 6 ABLATION STUDY

**Effect of Domain Descriptors.** Using both mean and variance to compute distributional similarity via KL divergence is theoretically well-founded, as these first- and second-order statistics jointly capture the essential characteristics of a distribution. As shown in Table 7, IMSE-Retrieval (Mean) and IMSE-Retrieval (Var) denote the variants of our method that retrieve spectral codes based on the mean and variance statistics, respectively. Retrieval with variance performs slightly better than IMSE-Retrieval (Mean), and the superior performance of IMSE-Retrieval, which combines both statistics, confirms their complementary roles.

**Component Ablation Studies.** We conduct two complementary ablation studies that separately analyze (1) the role of entropy minimization and diversity maximization under TTA setting, and (2) the impact of the diversity maximization and Domain Bank under CTTA setting, when entropy minimization is fixed. We estimate the individual contributions of entropy minimization and diversity

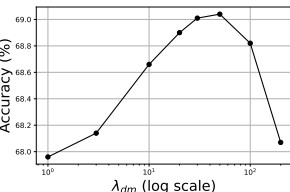

Figure 3: Hyperparameter sensitivity of $\lambda_{\mathrm{dm}}$.

Table 10: **Effect of target modules** of IMSE. ✓indicates the component is used.

| Attn.qkv | Attn.proj | MLP.fc1 | MLP.fc2 | Accuracy (%) |
|---|---|---|---|---|
| - | - | - | - | 51.0 |
| ✓ | - | - | - | 66.2 |
| - | ✓ | - | - | 66.8 |
| - | - | ✓ | - | 65.8 |
| - | - | - | ✓ | 67.2 |
| - | ✓ | - | ✓ | 68.6 |
| ✓ | ✓ | - | ✓ | 68.9 |
| ✓ | ✓ | ✓ | ✓ | **69.0** |

maximization. As shown in Table 8, $\mathcal{L}_{\mathrm{entmin}}$ provides the essential adaptation signal, while $\mathcal{L}_{\mathrm{dm}}$ prevents representation collapse during adaptation. This observation indicates that the two losses play complementary roles in TTA.

Next, we fix $\mathcal{L}_{\mathrm{entmin}}$ and further ablate the remaining two components: the diversity maximization loss and the Domain Bank–based retrieval mechanism. As shown in Table 9, when combined with entropy minimization, incorporating diversity maximization and the Domain Bank individually improves performance by 3.8 pp and 4.2 pp, respectively, demonstrating that each component is effective in the CTTA scenario. Incorporating all three components yields the highest performance, indicating that they play complementary roles in CTTA.

**Finetuning different layers.** To understand which ViT modules benefit most from fine-tuning singular values, we perform an ablation study in the TTA setting. As shown in Table 10, adapting the second MLP module (MLP.fc2) slightly outperforms the attention projection module (Attn.proj). Adapting both yields a further gain, indicating their complementary relationship. The best performance is achieved when all linear layers are adapted. For efficiency, adapting only the attention projection and MLP output layers provides a favorable trade-off.

**Hyperparameter sensitivity of $\lambda_{\mathrm{dm}}$.** We conduct an ablation study on the weight of the diversity maximization loss in the TTA setting. Using a supervised ViT-Base model, we change the weight over $[1, 3, 10, 20, 30, 50, 100, 200]$ and measure the average accuracy across the 15 common corruptions of ImageNet-C. As shown in Figure 3, we obtain the best accuracy of 69.0% when the weight is set to 50, and higher weights reduce accuracy. Notably, the method remains robust even as the scale of $\lambda_{\mathrm{dm}}$ varies widely from 1 to 200, and a weight of 200 still achieves around 68% accuracy.

## 7 CONCLUSION

In this paper, we propose IMSE, a parameter-efficient test-time adaptation (TTA) framework that adapts singular values via singular value decomposition while mitigating feature diversity collapse through a diversity maximization loss. For continual test-time adaptation (CTTA), we further introduce IMSE-Retrieval, which enables rapid adaptation by reusing previously adapted singular values based on domain similarity. Extensive experiments on ImageNet-C/R/A demonstrate that IMSE consistently improves performance across diverse pretraining strategies, including supervised, MAE, and CLIP pretraining. Furthermore, IMSE-Retrieval achieves state-of-the-art results in both CTTA and Gradual CTTA settings with up to 385 times fewer trainable parameters. We hope these results provide a strong foundation for future work on efficient and robust adaptation in continuously evolving real-world environments.

**Limitations.** While our method requires far fewer trainable parameters than existing approaches, it introduces some additional memory usage due to storing decomposed singular vectors alongside the original weights. In addition, the domain bank stores both the domain descriptors and the domain-specific adapted singular values, which requires extra storage in CTTA.

**Broader impact.** Our method enables efficient, robust adaptation of vision models under distribution shift, making it broadly applicable to real-world scenarios where test distribution differs from training distribution. However, care is needed when deploying adaptive systems without control over domain changes or failure detection.

## REPRODUCIBILITY STATEMENT

All experiments are conducted on publicly available datasets (e.g., ImageNet-C) without private data. Implementation builds on the publicly released DPAL and ViDA codebases, and all modifications necessary to reproduce our method are described in the manuscript. Section 4.1 and Appendix A include model architectures, hyper-parameters, optimization settings, and data-processing steps.

## ACKNOWLEDGMENTS

This research was supported by LG Energy Solution Co., Ltd.

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

## A    IMPLEMENTATION DETAILS

**Single-domain test-time adaptation (TTA).**    We use a batch size of 64 for all experiments. We employ Adam as an optimizer with Sharpness-Aware Minimization (SAM) (Foret et al., 2021). We apply a learning rate of 3e-3 for ImageNet-C (Hendrycks & Dietterich, 2019) (Hendrycks et al., 2021a) and ImageNet-R, and 4e-3 for ImageNet-A (Hendrycks et al., 2021b). Results of Supervised ViT-Base are taken from the DPAL (Tang et al., 2024). We exclude the final three transformer blocks of ViT from training, following the protocol established by SAR and DPAL.

Table 11: Learning rates for each method across different pretrained models.

| Method | MAE | CLIP | ViT-Large |
|--------|-----|------|-----------|
| TENT | 1e-3 | 1e-3 | 3e-4 |
| SAR | 1e-3 | 3e-3 | 1e-3 |
| DPAL | 4e-3 | 3e-3 | 4e-3 |
| IMSE | 1e-3 | 3e-3 | 4e-3 |

**CTTA and Gradual CTTA.**    We use the same batch size of 64 and adopt Adam as optimizer with SAM. We implement all baseline methods using their default configurations, following ViDA (Liu et al., 2024). In our method, we exclude the final three transformer blocks of ViT from training. We set the learning rate of IMSE-Retrieval to 7e-3 in CTTA, and 3e-3 in Gradual CTTA. The EMA coefficient used for updating the domain descriptor, $\alpha$, is set to 0.8. We use a domain-change threshold $\tau$ of 0.05 in CTTA and 0.02 in Gradual CTTA.

**Entropy-based sample filtering.**    We follow SAR (Niu et al., 2023)'s entropy filtering strategy in both scenarios, using the threshold $\tau = 0.4 \cdot \log(\# \text{ of classes})$ to exclude unreliable samples during adaptation.

**Diversity Maximization Loss.**    To preserve the model's class-discrimination ability, we introduce the diversity maximization loss $\mathcal{L}_{\text{dm}}$, which promotes diverse response patterns across spectral experts of each layer. We set the loss weight to $\lambda_{\text{dm}} = 50$ and apply $\mathcal{L}_{\text{dm}}$ to the set $\Lambda_{\text{dm}}$, which consists of all linear layers in the last three Transformer blocks.

## B    EXTENSION TO VIT-LARGE

In addition to the results in Section 4.3, we further evaluate our method on a larger architecture, ViT-Large, to examine its scalability and generalization. ViT-Large provides a significantly greater capacity for representation and is expected to leverage richer pretrained knowledge more effectively during the adaptation process. In Table 12, IMSE shows state-of-the-art performance compared to baseline methods. Our method outperforms SAR, the second-best performing approach, by approximately 4.1 pp. It demonstrates the substantial benefits of our spectral expert adaptation when applied to larger model architectures. These findings align with the core philosophy of our work—that our spectral expert adaptation method scales effectively with model capacity, enabling better utilization of the rich knowledge contained in more powerful pretrained models.

## C    ADDITIONAL ABLATION ON MODULE COMBINATION

We extend the ablation study in Section 6 of the main paper to a broader set of module combinations. We decompose the qkv projection in the attention module into three separate linear layers for q, k, and v. In Table 13, Attn.q, Attn.k, and Attn.v denote the individually separated components. As shown in Table 13, attention value (Attn.v), attention projection (Attn.proj), and second layer of MLP block (MLP.fc2) contribute substantially to adaptation performance. Notably, we find that explicitly separating the q, k, and v projections leads to further performance improvements, achieving a 0.2 pp gain compared to using the combined qkv projection.

Table 12: **Test-time adaptation on ImageNet-C (50k)**. Accuracy (↑) over 15 corruptions at severity level 5 using supervised pretrained ViT-Large model. Results with * are reproduced by us.

| | Noise | | | Blur | | | | Weather | | | | Digital | | | | |
|---|---|---|---|---|---|---|---|---|---|---|---|---|---|---|---|---|
| ImageNet-C (50k) | Gauss. | Shot | Impul. | Defoc. | Glass | Motion | Zoom | Snow | Frost | Fog | Brit. | Contr. | Elastic | Pixel. | JPEG | Avg.(↑) |
| Source | 62.5 | 62.0 | 63.3 | 52.9 | 45.3 | 60.7 | 55.2 | 66.0 | 62.4 | 62.6 | 79.9 | 40.1 | 56.2 | 74.3 | 72.8 | 61.1 |
| TENT* | 65.2 | 66.0 | 66.1 | 60.8 | 54.4 | 64.1 | 60.4 | 68.2 | 63.7 | 65.9 | 80.4 | 58.0 | 60.3 | 75.9 | 73.6 | 65.5 |
| SAR* | 67.4 | 68.3 | 69.3 | 55.8 | 59.6 | 66.1 | 64.2 | 68.8 | 68.1 | 70.5 | 81.6 | 59.0 | 69.7 | 78.0 | 75.8 | 68.1 |
| DPAL* | 66.6 | 68.4 | 68.1 | 62.5 | 63.8 | 67.9 | 65.7 | 72.0 | 64.4 | 69.5 | 81.5 | 56.5 | 71.3 | 73.6 | 75.7 | 68.5 |
| IMSE | **70.1** | **71.0** | **71.0** | **68.1** | **68.5** | **71.8** | **71.7** | **75.8** | **74.0** | **76.3** | **82.7** | **68.5** | **76.0** | **80.4** | **78.3** | **73.6** |

Table 13: **Ablation study on target modules.** The best is **bolded** and the second best is underlined.

| Attn.q | Attn.k | Attn.v | Attn.proj | MLP.fc1 | MLP.fc2 | Acc (↑) |
|---|---|---|---|---|---|---|
| - | - | - | - | - | - | 51.0 |
| ✓ | - | - | - | - | - | 59.6 |
| - | ✓ | - | - | - | - | 61.8 |
| - | - | ✓ | - | - | - | 66.6 |
| ✓ | ✓ | ✓ | - | - | - | 67.4 |
| ✓ | ✓ | ✓ | ✓ | - | - | 68.4 |
| ✓ | ✓ | ✓ | - | - | ✓ | 68.6 |
| ✓ | ✓ | ✓ | ✓ | - | ✓ | 69.0 |
| ✓ | ✓ | ✓ | ✓ | ✓ | ✓ | **69.2** |

# D    DIVERSITY ANALYSIS ACROSS TRANSFORMER BLOCKS

We analyze how the diversity of alignment statistics varies across network depth. Specifically, we measure the diversity at the second layer of the MLP block (MLP.fc2) within the 3rd, 6th, and 9th Transformer blocks. Figure 4 compares three settings: entropy–minimization loss ($\mathcal{L}_{\text{entmin}}$), our full objective ($\mathcal{L}_{\text{IMSE}}$), and cross-entropy loss ($\mathcal{L}_{\text{CE}}$).

The results show that when only $\mathcal{L}_{\text{entmin}}$ is applied, the diversity of expert-input alignments decreases more severely in deeper layers and appears noisier in the early layers. We assume that early layers are relatively less affected by the entropy-minimization loss and focus on extracting domain-related features rather than utilizing them. Based on these observations, we compute the diversity-maximization loss on the deeper Transformer blocks to better preserve feature diversity during adaptation.

# E    DOMAIN DESCRIPTOR SIMILARITY ANALYSIS

To better understand the behavior of domain descriptors across different test domains, we measure the pairwise distance between descriptors. In Figure 5, domains belonging to the noise category (e.g., Gaussian noise, Impulse noise, Shot noise) exhibit highly similar descriptors due to their shared low-level statistical characteristics. In contrast, the Contrast corruption shows significantly larger distances from all other domains in the descriptor space. This indicates that it possesses unique distributional properties.

# F    THE USAGE OF LARGE LANGUAGE MODELS

We utilize large language models (LLMs) as auxiliary tools during our experiments. We employ LLMs for debugging and writing repetitive code, and they were used for basic grammar correction during writing our manuscript. The research ideas, experimental design, and all scientific conclusions are our own work.

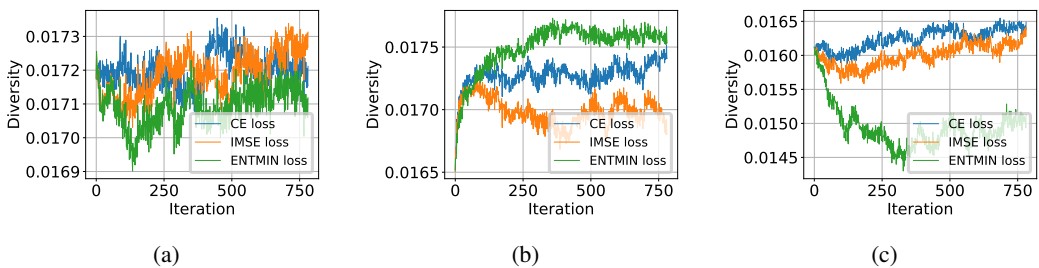

(a)                                            (b)                                            (c)

Figure 4: (a) Diversity of alignment patterns in 3rd Transformer Block. (b) Diversity of alignment patterns in 6th Transformer Block. (c) Diversity of alignment patterns in 9th Transformer Block.

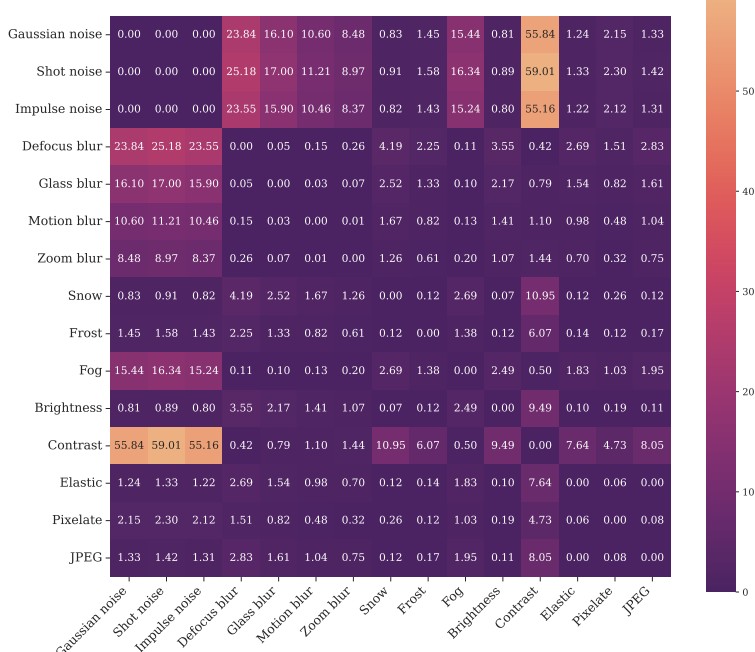

Figure 5: **Domain distance matrix.** Pairwise distance matrix among 15 domain descriptors.

Table 14: **Test-time adaptation on ImageNet-3DCC (50k)**. Accuracy(%)(↑) over 11 corruptions at severity level 5 using supervised pretraining ViT-Base. Results with * are reproduced by us.

| Method | bit error | color quantization | far focus | flash | fog | h265 abr | h265 crf | iso noise | low light | near focus | xy motion blur | z motion blur | Avg.(↑) |
|---|---|---|---|---|---|---|---|---|---|---|---|---|---|
| Source* | **20.3** | 53.1 | 61.7 | 47.3 | 39.4 | 52.8 | 57.3 | 59.1 | 56.3 | 70.5 | 43.2 | 47.4 | 50.7 |
| TENT* | 1.2 | 64.1 | 68.2 | 52.1 | 7.5 | 56.1 | 60.7 | 65.6 | 70.3 | 76.2 | 56.4 | 61.2 | 53.3 |
| SAR* | 13.8 | 64.7 | 68.7 | 54.3 | 41.4 | 57.1 | 61.4 | 66.3 | 70.8 | 76.4 | 57.8 | 62.5 | 57.9 |
| DPAL* | 13.1 | 64.2 | 68.3 | 53.7 | 43.4 | 57.0 | 61.2 | 65.8 | 70.0 | 75.8 | 56.6 | 61.7 | 57.6 |
| IMSE | 15.5 | **65.0** | **70.1** | **55.2** | **44.5** | **58.1** | **62.2** | **67.3** | **71.6** | **77.1** | **58.9** | **63.9** | **59.1** |

Table 15: **Test-time adaptation on OfficeHome**. Accuracy(%)(↑) over 12 source–target domain pairs and average performance. Results with * are reproduced by us.

| Method | R→A | R→C | R→P | A→R | A→C | A→P | C→R | C→A | C→P | P→R | P→A | P→C | Avg.(↑) |
|---|---|---|---|---|---|---|---|---|---|---|---|---|---|
| Source* | 79.68 | 56.05 | 88.35 | 87.46 | 58.55 | 82.90 | 84.66 | 78.37 | 84.29 | 87.74 | 75.32 | 55.69 | 76.58 |
| TENT* | 79.68 | 56.08 | **88.91** | 88.01 | 59.61 | 83.10 | 85.42 | **79.68** | **85.01** | 87.90 | 75.48 | 55.71 | 77.05 |
| SAR* | 79.72 | **56.72** | 88.37 | 87.62 | **61.42** | 83.03 | 85.40 | 79.48 | 84.61 | 87.81 | 76.02 | 56.26 | 77.20 |
| DPAL* | 79.93 | 55.57 | 86.52 | 87.14 | 59.17 | 82.61 | 85.33 | 79.02 | 84.09 | 86.68 | 75.85 | 55.25 | 76.43 |
| IMSE | **81.04** | 56.35 | **88.91** | **88.13** | 60.36 | **83.44** | **85.74** | 79.39 | 84.88 | **88.04** | **76.14** | **56.28** | **77.39** |

# G ADDITIONAL RESULTS

**Challenging benchmarks.** We further evaluate IMSE under more challenging domain-shift benchmark, ImageNet-3DCC. Compared to standard 2D common corruptions (Hendrycks & Dietterich, 2019), ImageNet-3DCC (Kar et al., 2022) is more challenging as its corruptions are generated using 3D scene geometry, resulting in more realistic and real-world–plausible distribution shifts. All experiments are conducted in a test-time adaptation setting using a supervised pretrained ViT-Base model. As shown in Table 14, IMSE achieves consistently strong performance across all corruptions, demonstrating its effectiveness under challenging domain-shift benchmarks.

**Domain Adaptation Benchmarks.** We additionally evaluate our method on the domain adaptation benchmarks, Office-Home (Venkateswara et al., 2017) and DomainNet (Peng et al., 2019) under test-time adaptation setting (using supervised pretrained ViT-Base). In our experiments, we use 126 categories from DomainNet, selecting Real, Clipart, Painting, and Sketch as the evaluation domains following MME (Saito et al., 2019). For OfficeHome, we use all 65 categories and include Real World, Art, Clipart, and Product as domains. Domain adaptation benchmarks differ from corruption-based benchmarks because they require additional supervised training and exhibit different class distributions across domains. Despite these differences, IMSE achieves consistently strong performance on both benchmarks in Table 15, Table 16. This demonstrates that IMSE is effective not only on corruption-based TTA benchmarks but also on domain adaptation benchmarks.

# H LONG-TERM CONTINUAL TEST-TIME ADAPTATION

We evaluate the long-term stability and scalability of IMSE-Retrieval. To simulate long-term continual test-time adaptation setting where the same domain reappears multiple times, we split each ImageNet-C corruption (all at severity 5) into 10 datasets of different 5,000 images. This setup reflects realistic deployment scenarios where the same domain recurs multiple times with different samples. As shown in Table 17, the proposed method maintains strong and stable performance across all rounds and consistently outperforms prior approaches. These results demonstrate that IMSE-Retrieval is robust under repeated re-occurrence of the same domain.

Table 16: **Test-time adaptation on DomainNet**. Accuracy(%)(↑) over 12 source–target domain pairs and average performance. Results with * are reproduced by us.

| Method | R→C | R→P | R→S | C→R | C→P | C→S | P→R | P→C | P→S | S→R | S→C | S→P | Avg.(↑) |
|---|---|---|---|---|---|---|---|---|---|---|---|---|---|
| Source[*] | 66.84 | 76.14 | 59.44 | 80.46 | 69.85 | 65.93 | 86.72 | 69.04 | **59.95** | 82.19 | 72.95 | **73.62** | 71.93 |
| TENT[*] | 68.25 | 76.45 | 60.36 | _81.91_ | 72.32 | 67.01 | _87.60_ | 69.97 | 59.21 | 82.76 | 73.05 | 73.11 | 72.67 |
| SAR[*] | **69.64** | **77.78** | **64.23** | _81.91_ | _72.83_ | **67.73** | 86.72 | **70.77** | 59.41 | _82.77_ | _73.53_ | 73.41 | _73.39_ |
| DPAL[*] | 68.14 | 77.54 | 60.86 | 81.78 | 72.79 | 66.07 | 84.93 | 70.16 | 58.86 | 81.61 | 72.54 | 70.11 | 72.12 |
| IMSE | _69.33_ | **77.78** | _62.40_ | **82.45** | **73.23** | _67.57_ | **87.85** | _70.42_ | _59.72_ | **83.10** | **73.73** | _73.55_ | **73.43** |

Table 17: **Continual test-time adaptation on ImageNet-C under the domain-recurring setting.** Average accuracy (%) for each round (R1–R10) and the average over all rounds. Results with * are reproduced by us.

| Method | R1 | R2 | R3 | R4 | R5 | R6 | R7 | R8 | R9 | R10 | Avg.(↑) |
|---|---|---|---|---|---|---|---|---|---|---|---|
| Source[*] | 49.2 | 50.2 | 50.4 | 50.0 | 49.8 | 50.4 | 50.7 | 49.8 | 50.4 | 50.0 | 50.1 |
| TENT[*] | 53.3 | 56.9 | 57.8 | 57.8 | 57.5 | 58.7 | 59.2 | 57.9 | 57.9 | 58.1 | 57.5 |
| CoTTA[*] | 49.7 | 52.1 | 53.1 | 53.1 | 53.2 | 54.0 | 54.6 | 53.5 | 54.0 | 54.0 | 53.1 |
| ViDA[*] | _53.7_ | _57.2_ | _59.3_ | _58.7_ | _59.1_ | _59.5_ | _59.4_ | _59.7_ | _59.6_ | _59.8_ | _58.6_ |
| IMSE-Retrieval | **63.7** | **65.1** | **65.8** | **65.2** | **64.8** | **65.9** | **65.0** | **65.4** | **65.1** | **65.3** | **65.1** |

