# OpenReview forum: "IMSE: Intrinsic Mixture of Spectral Experts Fine-tuning for Test-Time Adaptation"
_ICLR.cc/2026/Conference — ICLR 2026 Poster_

### Official Review · Reviewer_Q6jF · 2025-10-29

**Soundness:** 2
**Presentation:** 3
**Contribution:** 3
**Rating:** 4
**Confidence:** 4

**Summary:**

The paper introduces IMSE (Intrinsic Mixture of Supervised Experts), a test-time adaptation method that decomposes linear layers in a pretrained model using singular value decomposition (SVD), treating each rank-1 component as a spectral expert. During adaptation, only the singular values are updated while the bases remain fixed, allowing parameter-efficient updates. The method combines entropy minimization with confidence-based filtering, a diversity maximization loss to promote balanced expert activation, and a domain descriptor mechanism for detecting distribution shifts and retrieving previously adapted spectral parameters from a memory bank.

**Strengths:**

The use of spectral decomposition to form a mixture-of-experts representation is an interesting approach to test-time adaptation.

**Weaknesses:**

1. The assumption that spectral experts can generalize across domains by only adapting singular values lacks theoretical justification and is not thoroughly evaluated in challenging domain-shift scenarios.
2. The interplay between the different components (e.g., diversity loss, entropy minimization, and domain memory) is complex, but the paper lacks sufficient ablation studies to isolate their individual contributions.
3. The scalability and stability of the domain memory mechanism under long-term continual adaptation are not fully explored, leaving questions about its robustness in practice.

**Questions:**

1. he framework assumes that adapting only the singular values of pretrained linear layers is sufficient for capturing domain shifts. Could the authors provide theoretical or empirical evidence supporting this assumption, especially for shifts that may require new basis directions?
2. The method combines entropy minimization, diversity regularization, and domain memory. Have the authors performed detailed ablations to assess the individual and joint impact of each component? For example, how does performance change when the diversity loss is removed?
3. The KL-divergence-based domain descriptor mechanism is used to trigger new adaptations and retrieve stored parameters. How robust is this approach under gradual or noisy distribution shifts? What mechanisms are in place to prevent over-segmentation or memory bloat when many small shifts occur?\
4. The diversity regularization loss penalizes widespread activation of individual spectral experts. Could this unintentionally suppress task-relevant features that happen to be common in a new domain? How does the method balance between promoting diversity and retaining important features?
5. Could the authors discuss whether the IMSE framework or its components (e.g., diversity loss, spectral adaptation) can be extended to multi-source domain generalization or continual learning settings where test-time updates are not allowed?
6. While the method is evaluated on common TTA benchmarks, it would be helpful to include results on tasks involving more gradual or subtle domain shifts, or with more fine-grained domain boundaries. Could the authors include such evaluations in a revision?

---

> ### Author Response · Authors · 2025-11-21
>
> We thank the reviewer for the detailed and constructive feedback. The concerns you raised help us refine the presentation and strengthen the empirical evaluation. We address each point sequentially below, and we incorporate the corresponding clarifications and additional results in the revised manuscript.
>
> ---
>
> ### **W1 & Q1 - Design Justification and Empirical Evidence**
>
> To address this expressivity concern, we provide both a design justification and empirical evidence as follows:
>
> **Design Justification**
>
> Our method interprets each linear layer as a mixture of spectral experts and adapts to the test domain by updating only the singular values, while keeping the spectral bases fixed. This design assumes that the pretrained model already provides sufficiently expressive feature extractors. Updating spectral bases increase both trainable parameters and Domain Bank storage, which is undesirable in CTTA.
>
> **Empirical Evidence**
>
> To investigate whether our method remains effective without adjusting spectral directions, we further evaluate IMSE under more challenging domain-shift benchmarks. We evaluate IMSE on ImageNet-V2 (distribution shift), Sketch (texture shift), 3DCC (geometry-aware real-world corruptions), and OfficeHome & DomainNet(cross-domain shift). Specifically, ImageNet-3DCC [1] is more challenging than standard 2D common corruptions [2]. Its corruptions are generated using 3D scene geometry, resulting in more realistic and real-world–plausible distribution shifts. As shown in Tables 4-A, 4-B, 4-C, and 4-D, **IMSE consistently shows strong performance across all benchmarks.** These results demonstrate that singular-value adaptation alone is sufficiently expressive to handle diverse and challenging domain shifts.
>
> Table 4-A. Test-time adaptation on ImageNet-V2/Sketch
>
> |  | ImageNet-V2 | ImageNet-Sketch |
> | --- | --- | --- |
> | Source | 73.83 | 43 |
> | TENT | 74.28 | 50.13 |
> | SAR | 74.35 | 52.1 |
> | DPAL | 74.4 | 52.25 |
> | IMSE | **74.48** | **53.58** |
>
> Table 4-B. Test-time adaptation on ImageNet-3DCC, accuracy(%) across 11 corruption domains and the average performance.
>
> | Method | bit error | color quantization | far focus | flash | fog | h265 abr | h265 crf | iso noise | low light | near focus | xy motion blur | z motion blur | Avg. |
> | --- | --- | --- | --- | --- | --- | --- | --- | --- | --- | --- | --- | --- | --- |
> | Source | **20.3** | 53.1 | 61.7 | 47.3 | 39.4 | 52.8 | 57.3 | 59.1 | 56.3 | 70.5 | 43.2 | 47.4 | 50.7 |
> | TENT | 1.2 | 64.1 | 68.2 | 52.1 | 7.5 | 56.1 | 60.7 | 65.6 | 70.3 | 76.2 | 56.4 | 61.2 | 53.3 |
> | SAR | 13.8 | 64.7 | 68.7 | 54.3 | 41.4 | 57.1 | 61.4 | 66.3 | 70.8 | 76.4 | 57.8 | 62.5 | 57.9 |
> | DPAL | 13.1 | 64.2 | 68.3 | 53.7 | 43.4 | 57.0 | 61.2 | 65.8 | 70.0 | 75.8 | 56.6 | 61.7 | 57.6 |
> | IMSE | 15.5 | **65.0** | **70.1** | **55.2** | **44.5** | **58.1** | **62.2** | **67.3** | **71.6** | **77.1** | **58.9** | **63.9** | **59.1** |
>
> Table 4-C. Test-time adaptation on OfficeHome, accuracy across 12 source and target domain pairs and average performance.
>
> |  | R->A | R-C | R->P | A->R | A->C | A->P | C->R | C->A | C->P | P->R | P->A | P->C | Avg. |
> | --- | --- | --- | --- | --- | --- | --- | --- | --- | --- | --- | --- | --- | --- |
> | Source | 79.68 | 56.05 | 88.35 | 87.46 | 58.55 | 82.9 | 84.66 | 78.37 | 84.29 | 87.74 | 75.32 | 55.69 | 76.58 |
> | TENT | 79.68 | 56.08 | **88.91** | 88.01 | 59.61 | 83.1 | 85.42 | **79.68** | **85.01** | 87.9 | 75.48 | 55.71 | 77.05 |
> | SAR | 79.72 | **56.72** | 88.37 | 87.62 | **61.42** | 83.03 | 85.4 | 79.48 | 84.61 | 87.81 | 76.02 | 56.26 | 77.20 |
> | DPAL | 79.93 | 55.57 | 86.52 | 87.14 | 59.17 | 82.61 | 85.33 | 79.02 | 84.09 | 86.68 | 75.85 | 55.25 | 76.43 |
> | IMSE | **81.04** | 56.35 | **88.91** | **88.13** | 60.36 | **83.44** | **85.74** | 79.39 | 84.88 | **88.04** | **76.14** | **56.28** | **77.39** |
>
> Table 4-D. Test-time adaptation on DomainNet, accuracy across 12 source and target domain pairs and average performance.
>
> |  | R->C | R->P | R->S | C->R | C->P | C->S | P->R | P->C | P->S | S->R | S->C | S->P | Avg. |
> | --- | --- | --- | --- | --- | --- | --- | --- | --- | --- | --- | --- | --- | --- |
> | Source | 66.84 | 76.14 | 59.44 | 80.46 | 69.85 | 65.93 | 86.72 | 69.04 | **59.95** | 82.19 | 72.95 | **73.62** | 71.93 |
> | TENT | 68.25 | 76.45 | 60.36 | 81.91 | 72.32 | 67.01 | 87.6 | 69.97 | 59.21 | 82.76 | 73.05 | 73.11 | 72.67 |
> | SAR | **69.64** | **77.78** | **64.23** | 81.91 | 72.83 | **67.73** | 86.72 | **70.77** | 59.41 | 82.77 | 73.53 | 73.41 | 73.39 |
> | DPAL | 68.14 | 77.54 | 60.86 | 81.78 | 72.79 | 66.07 | 84.93 | 70.16 | 58.86 | 81.61 | 72.54 | 70.11 | 72.12 |
> | IMSE | 69.33 | **77.78** | 62.4 | **82.45** | **73.23** | 67.57 | **87.85** | 70.42 | 59.72 | **83.1** | **73.73** | 73.55 | **73.43** |

---

> ### Author Response · Authors · 2025-11-21
>
> ### **W2 & Q2 - Component Ablation Studies**
>
> We present two complementary ablation studies that separately analyze (1) the role of entropy minimization and diversity maximization under various pretraining strategies, and (2) the additional component ablation study of the diversity loss and Domain Bank when entropy minimization is fixed.
>
> **Impact of Entropy minimization loss.**
> First, we analyze the loss functions across different pretraining strategies. As shown in Table 4-E, entropy minimization provides the essential adaptation signal. provides the main adaptation signal, while diversity maximization prevents representation collapse during this adaptation process, making the two losses complementary in TTA.
>
> Table 4-E. Test-time adaptation ablation study on loss components using ImageNet-C.
>
> | Pretraining (ViT-B) | $\mathcal{L}_\text{entmin}$ | $\mathcal{L}_\text{dm}$ | Average Accuracy. |
> | --- | --- | --- | --- |
> | Supervised | O | X | 67.8 |
> | Supervised | X | O | 32.7 |
> | Supervised | O | O | **69.1** |
>
> **Impact of Diversity maximization loss and Domain Bank.**
>
> Next, we fix entropy minimization loss and further ablate the remaining two components: $\mathcal{L}_\text{dm}$ and the domain bank–based retrieval mechanism. Table 4-F demonstrates that incorporating all three components yields the highest performance, indicating that they play complementary roles in CTTA.
>
> Table 4-F. Continual test-time adaptation ablation study on loss components using ImageNet-C.
>
> | $\mathcal{L}_\text{entmin}$ | $\mathcal{L}_\text{dm}$ | Domain Bank | Average Accuracy. |
> | --- | --- | --- | --- |
> | O | X | X | 59.4 |
> | O | O | X | 62.2 |
> | O | X | O | 62.8 |
> | O | O | O | **64.4** |
>
> ---
>
> ### **Q3 - KL-divergence based Domain Descriptor Mechanism under Gradual Shifts**
>
> To evaluate whether the proposed method can reliably discriminate gradual domain shifts, we conduct an additional **gradual shift** experiment. Following the same domain order of Table 4 in the manuscript (e.g., Gaussian Noise (GN) → Shot Noise → Impulse Noise (IN) → Defocus Blur (DB) → … → JPEG Compression), we construct a sequence where the severity level gradually changes as 1→2→3→4→5→4→3→2→1 within each corruption type. With 15 corruption types repeated 9 times, this setting produces 135 sequential test domains. As shown in Table 4-F, **IMSE-Retrieval achieves the highest performance.**
>
> Table 4-F. Continual test-time adaptation on ImageNet-C under gradual shifts.
>
> | Method | Average accuracy. |
> | --- | --- |
> | Source | 67.3 |
> | TENT | 70.7 |
> | CoTTA | 69.5 |
> | ViDA | 72.5 |
> | IMSE-Retrieval | **74.9** |
>
> We additionally analyze its retrieval behavior using a domain change threshold of $\tau=0.2$, as detailed below.
>
> 1. **GN1→…→GN5 (increasing severity):** IMSE-Retrieval consistently retrieves the spectral code corresponding to the closest severity level.
> 2. **GN5→…→GN1 (decreasing severity):** IMSE-Retrieval  retrieves the encountered domains in the increasing stage.
> 3. **Transition from IN1 to DB1:** IMSE-Retrieval  correctly detects a domain change and selects the pretrained domain as the most similar domain.
> 4. **DB1→DB5→DB1:** Retrieval process remains stable and consistent, as observed in the preceding scenario
>
> These results suggest that the descriptor remains responsive under gradual distribution changes.
>
> Furthermore, to address the over-segmentation issue, we increase the domain-change threshold $\tau$ and observe how this affects both domain-change detection quality and adaptation performance. In evaluating domain-change detection, we allow a margin of one step around each transition point to account for the fine-grained nature of gradual shifts. When the threshold is set to a more tolerant value ($\tau = 0.04$), the F1 score of domain-change detection decreases F1: 0.77 **→** 0.28), while the overall performance increases (accuracy: 74.9 **→** 75.6). This trade-off indicates that sensitive domain change detection is not always advantageous, and that choosing a moderate threshold helps achieve strong performance while keeping the Domain Bank compact.

---

> ### Author Response · Authors · 2025-11-21
>
> ### **Q4 - Robustness of Diversity Maximization in Non-i.i.d. Settings**
>
> To address the concern that diversity maximization might suppress task-relevant features, we evaluate our method under extreme label-imbalanced scenarios where such suppression risks are highest. We acknowledge that if a batch contains samples from a single class (non-i.i.d.), encouraging widespread activation could theoretically conflict with preserving class-discriminative features. To investigate this, we conduct experiments using the Dirichlet non-i.i.d. setting (following DELTA [3]), varying the concentration factor $\alpha$ (1.0 $\to$ 0.1). A smaller $\alpha$ indicates a more severe class imbalance within each batch.
>
> As shown in Table 4-G, incorporating the diversity maximization loss consistently improves adaptation accuracy by approximately 1.1% across all degrees of non-i.i.d. shifts. Notably, even under the most severe class imbalance ($\alpha=0.05$), using diversity maximization loss boosts performance from 67.33% to 68.41%. This stable improvement confirms that $\mathcal{L}_\text{dm}$ does not suppress task-relevant features. Instead, it effectively complements the adaptation process, maintaining robust representations even when the input distribution is extremely skewed.
>
> Table 4-G.  Test-time adaptation on ImageNet-C under non-i.i.d setting, Average accuracy across 15 domains.
>
> | **Method** | **α=1.0 (Mild)** | **α=0.5 (Moderate)** | **α=0.1 (Severe)** | **α=0.05 (More Severe)** |
> | --- | --- | --- | --- | --- |
> | IMSE | **68.92** | **68.88** | **68.62** | **68.41** |
> | IMSE w.o $\mathcal{L}_\text{dm}$ | 67.79 | 67.78 | 67.64 | 67.33 |
> | Improvement | **+1.13** | **+1.1** | **+0.98** | **+1.08** |
>
> ---
>
> ### **Q5 - Potential Extensions of IMSE Beyond TTA**
>
> Although IMSE is designed for TTA, several components can conceptually extend to other problem settings. **First, the diversity maximization loss** reduces domain-specific feature collapse in the absence of labels. In settings like multi-source domain generalization or continual learning, where class labels provide strong supervision, its benefit would likely be limited. **Second, spectral adaptation** updates only the singular values while preserving pretrained bases. In continual learning, spectral adaptation might offer a better stability–plasticity trade-off, but its benefits may depend on the characteristics of the encountering tasks. **Third, the domain bank and retrieval mechanism** identifies domains online using distributional descriptors. A similar approach could be used in continual learning where task identities are unknown. However, because continual learning and TTA differ in training, evaluation, and benchmark characteristics, applying this mechanism would require redefining the task descriptor and adapting the retrieval strategy.
>
> ---
>
> ### **Q6 - Additional Evaluation on Gradual Domain Shifts**
>
> We additionally evaluate our method under a much more gradual and fine grained domain shift scenario, as detailed in our response to Q3. Table4-F shows that **IMSE-Retrieval outperforms other baselines** without over-segmentation. In this setting, IMSE-Retrieval detects gradual domain changes and retrieves appropriate domains. In response to the reviewer’s suggestion that evaluating more subtle shifts would strengthen the paper, we have incorporated the gradual shift experiment.

---

> > ### Comment · Reviewer_Q6jF · 2025-11-23
> >
> > The authors have addressed my earlier concerns, which has led me to raise my original score.

---

> > > ### Author Response · Authors · 2025-11-24
> > >
> > > We sincerely thank you for your thoughtful review and for adjusting your score. Your feedback has been instrumental in refining our work. We are glad that our response addressed your concerns, and your comments have helped strengthen the validity of our proposed method.

---

> ### Author Response · Authors · 2025-11-21
>
> ### W3 - Scalability and Stability of IMSE-Retrieval under Long-term CTTA
>
> We evaluate the long-term stability and scalability of the proposed domain memory mechanism through two complementary experiments.
>
> First, we have conducted a **gradual shift evaluation**, which produces a sequence roughly nine times longer than the standard CTTA setting. As detailed in our responses to Q3 and Q6, IMSE-Retrieval reliably detects domain changes and maintains consistent retrieval throughout this extended sequence.
>
> Second, to directly assess long-term continual adaptation, we perform a **domain-recurring evaluation** by splitting each ImageNet-C corruption (all at severity 5) into ten equal-sized domains of 5,000 images. This setup reflects realistic scenarios where the same domain reappears multiple times with different samples. As shown in Table 4-H, the proposed method maintains strong and stable performance, consistently outperforming prior approaches.
>
> These results demonstrate that **IMSE-Retrieval is robust in both long-term scenarios**: extended gradual shifts and repeated re-occurrence of the same domain.
>
> Table 4-H. Continual test-time adaptation on ImageNet-C under Domain recurring setting, average accuracy across 10 rounds and the average performance.
>
> | Name | R1 | R2 | R3 | R4 | R5 | R6 | R7 | R8 | R9 | R10 | Avg. |
> | --- | --- | --- | --- | --- | --- | --- | --- | --- | --- | --- | --- |
> | Source | 49.2 | 50.2 | 50.4 | 50.0 | 49.8 | 50.4 | 50.7 | 49.8 | 50.4 | 50.0 | 50.1 |
> | TENT | 53.3 | 56.9 | 57.8 | 57.8 | 57.5 | 58.7 | 59.2 | 57.9 | 57.9 | 58.1 | 57.5 |
> | CoTTA | 49.7 | 52.1 | 53.1 | 53.1 | 53.2 | 54.0 | 54.6 | 53.5 | 54.0 | 54.0 | 53.1 |
> | ViDA | 53.7 | 57.2 | 59.3 | 58.7 | 59.1 | 59.5 | 59.4 | 59.7 | 59.6 | 59.8 | 58.6 |
> | IMSE-Retrieval | **63.7** | **65.1** | **65.8** | **65.2** | **64.8** | **65.9** | **65.0** | **65.4** | **65.1** | **65.3** | **65.1** |
>
> ---
>
> ### **Reference**
>
> [1] Kar, Oğuzhan Fatih, et al. "3d common corruptions and data augmentation." *Proceedings of the IEEE/CVF Conference on Computer Vision and Pattern Recognition*. 2022.
>
> [2] Hendrycks, Dan, and Thomas Dietterich. "Benchmarking Neural Network Robustness to Common Corruptions and Perturbations." *International Conference on Learning Representations*.
>
> [3] Zhao, Bowen, Chen Chen, and Shu-Tao Xia. "DELTA: DEGRADATION-FREE FULLY TEST-TIME ADAPTATION." *The Eleventh International Conference on Learning Representations*.

---

### Official Review · Reviewer_wHid · 2025-10-30

**Soundness:** 2
**Presentation:** 3
**Contribution:** 3
**Rating:** 4
**Confidence:** 3

**Summary:**

This paper addresses the test-time adaptation (TTA) problem by leveraging the knowledge in pretrained networks. Specifically, it decomposes each linear layer via singular value decomposition (SVD) and adapts only the singular values. For continual test-time adaptation (CTTA), the method estimates input distributions to detect domain shifts and retrieves previously adapted singular values for rapid adaptation. Extensive experiments demonstrate that the proposed approach achieves state-of-the-art performance.

**Strengths:**

1. Using $\text{Std}_{i}^{(l)}$ to measure whether each expert captures domain-specific patterns is an interesting design.

2. Detecting new domains by comparing the current input-level descriptor $ϕ$ with the accumulated descriptor $\phi_{(t)}$ is an effective way.

3. This paper analyzes, in an unsupervised manner, that entropy minimization tends to capture domain-related rather than class-discriminative information.

**Weaknesses:**

1.  In lines 161–163, since $\mathbf{v}_i^{(l)}$ fixed, it is unclear how it can define each spectral expert’s response. Could the authors provide a more detailed explanation of this point.

2. The paper conducts experiments only on three OOD datasets. Could authors also evaluated the method on classical domain adaptation benchmarks such as Office-Home[1] or DomainNet[2]?

3. The paper introduces two loss components, entropy minimization $L_{entmin}$ and diversity maximization $L_{dm}$, but lacks an ablation study isolating their individual contributions. Could the author show the effect of using only entropy minimization, only diversity maximization, and their combination to better understand each term’s impact on adaptation performance.

4. The paper does not include experiments on time analysis (e.g., runtime efficiency) or memory analysis (e.g., computational cost and storage of spectral codes). Could authors provide such experiments to better demonstrate the efficiency of the proposed method?

[1] Venkateswara, Hemanth, et al. "Deep hashing network for unsupervised domain adaptation." Proceedings of the IEEE conference on computer vision and pattern recognition. 2017.

[2] Peng, Xingchao, et al. "Moment matching for multi-source domain adaptation." Proceedings of the IEEE/CVF international conference on computer vision. 2019.

**Questions:**

Please see the above weaknesses. If you can conduct additional experiments to further evaluate your method, I would be willing to raise my score.

---

> ### Author Response · Authors · 2025-11-21
>
> We thank the reviewer for the constructive feedback and for recognizing the strengths of our method. The reviewer raised several helpful points regarding the definition of spectral expert responses, evaluations on additional benchmarks, the contribution of each loss term, and the efficiency analysis. We address each concern below and provide additional experiments and clarifications.
>
> ---
>
> ### **W1 : Clarification of Spectral Expert’s Response**
>
> We clarify **how each spectral expert’s response is defined** and how the expert–input alignment is computed. Specifically, in the $l$-th layer, the $i$-th spectral expert corresponds to the rank-1 component $u_{(i)}^{(l)} \, \mu_{(i)}^{(l)} \, v_{(i)}^{(l)\top}$, as formulated in Eq (1) (Line 140). The output direction of each spectral expert is determined by the fixed left singular vector $u_{(i)}^{(l)}$, while its response strength to an input $x_n^{(l)}$ is given by the scalar $\mu_{(i)}^{(l)} \, v_{(i)}^{(l)\top} x_n^{(l)}$. To quantify how strongly each spectral direction is utilized, we normalize the projection $v_{(i)}^{(l)\top} x_n^{(l)}$ (excluding the singular value) and define this quantity as the expert–input alignment. We then compute statistics over these alignment values for each spectral expert, as described in Section 3.1.
>
> ---
>
> ### **W2 - Additional Results on DA Benchmarks**
>
> Following the reviewer’s suggestion, we additionally evaluate our method on the classical domain adaptation benchmarks, Office-Home and DomainNet under single test-time adaptation setting (using supervised pretrained ViT-Base). Domain adaptation benchmarks differ from corruption-based OOD benchmarks because they require additional supervised training and exhibit different class distributions across domains. Despite these differences, **IMSE achieves consistently strong performance on both benchmarks in Table 3-A and 3-B, outperforming existing TTA baselines.** This demonstrates that IMSE is effective not only on corruption-based TTA benchmarks but also on classical domain adaptation benchmarks.
>
> Table 3-A. Test-time adaptation on OfficeHome, accuracy across 12 source and target domain pairs and average performance.
>
> |  | R->A | R-C | R->P | A->R | A->C | A->P | C->R | C->A | C->P | P->R | P->A | P->C | Avg. |
> | --- | --- | --- | --- | --- | --- | --- | --- | --- | --- | --- | --- | --- | --- |
> | Source | 79.68 | 56.05 | 88.35 | 87.46 | 58.55 | 82.9 | 84.66 | 78.37 | 84.29 | 87.74 | 75.32 | 55.69 | 76.58 |
> | TENT | 79.68 | 56.08 | **88.91** | 88.01 | 59.61 | 83.1 | 85.42 | **79.68** | **85.01** | 87.9 | 75.48 | 55.71 | 77.05 |
> | SAR | 79.72 | **56.72** | 88.37 | 87.62 | **61.42** | 83.03 | 85.4 | 79.48 | 84.61 | 87.81 | 76.02 | 56.26 | 77.20 |
> | DPAL | 79.93 | 55.57 | 86.52 | 87.14 | 59.17 | 82.61 | 85.33 | 79.02 | 84.09 | 86.68 | 75.85 | 55.25 | 76.43 |
> | IMSE | **81.04** | 56.35 | **88.91** | **88.13** | 60.36 | **83.44** | **85.74** | 79.39 | 84.88 | **88.04** | **76.14** | **56.28** | **77.39** |
>
> Table 3-B. Test-time adaptation on DomainNet, accuracy across 12 source and target domain pairs and average performance.
>
> |  | R->C | R->P | R->S | C->R | C->P | C->S | P->R | P->C | P->S | S->R | S->C | S->P | Avg. |
> | --- | --- | --- | --- | --- | --- | --- | --- | --- | --- | --- | --- | --- | --- |
> | Source | 66.84 | 76.14 | 59.44 | 80.46 | 69.85 | 65.93 | 86.72 | 69.04 | **59.95** | 82.19 | 72.95 | **73.62** | 71.93 |
> | TENT | 68.25 | 76.45 | 60.36 | 81.91 | 72.32 | 67.01 | 87.6 | 69.97 | 59.21 | 82.76 | 73.05 | 73.11 | 72.67 |
> | SAR | **69.64** | **77.78** | **64.23** | 81.91 | 72.83 | **67.73** | 86.72 | **70.77** | 59.41 | 82.77 | 73.53 | 73.41 | 73.39 |
> | DPAL | 68.14 | 77.54 | 60.86 | 81.78 | 72.79 | 66.07 | 84.93 | 70.16 | 58.86 | 81.61 | 72.54 | 70.11 | 72.12 |
> | IMSE | 69.33 | **77.78** | 62.4 | **82.45** | **73.23** | 67.57 | **87.85** | 70.42 | 59.72 | **83.1** | **73.73** | 73.55 | **73.43** |

---

> ### Author Response · Authors · 2025-11-21
>
> ### **W3 : Component Ablation Study**
>
> We conduct an ablation study to isolate the individual contributions of entropy minimization and diversity maximization. Table 3-C shows that using $\mathcal{L}_\text{dm}$ alone leads to a substantial performance drop, while combining it with entropy minimization loss yields the best adaptation performance. Entropy minimization provides the main adaptation signal. In addition, diversity maximization prevents representation collapse during this process, making the two losses complementary in TTA.
>
>
> Table 3-C. Test-time adaptation ablation study on loss components using ImageNet-C.
>
> | Pretraining (ViT-B) | $\mathcal{L}_\text{entmin}$ | $\mathcal{L}_\text{dm}$ | Average Accuracy |
> | --- | --- | --- | --- |
> | Supervised | O | X | 67.8 |
> | Supervised | X | O | 32.7 |
> | Supervised | O | O | **69.1** |
>
> ---
>
> ### **W4 : Efficiency of IMSE-Retrieval**
>
> We measure the runtime efficiency and the additional storage of IMSE-Retrieval. As shown in Table 3-D, we report the per-batch processing time (batch size = 64) using a A6000 GPU. First, IMSE-Retrieval runs **3.5 times faster than ViDA and 2.5 times faster than CoTTA**. While its runtime is higher than the lightweight Tent baseline, IMSE delivers significantly better accuracy, resulting in a more favorable performance–runtime trade-off. Additionally, the diversity maximization loss introduces only a marginal overhead of 0.15 seconds. Second, the additional storage required is approximately **0.33 MB per domain.** Even with 100 stored domains, the overhead remains negligible relative to the backbone model size (330.23MB). These results show that the proposed approach not only achieves strong adaptation performance in CTTA scenarios but also maintains high computational and memory efficiency.
>
> Table 3-D. CTTA Runtime comparision (Batch Size = 64)
>
> |  | Source | TENT | ViDA | CoTTA | IMSE-Retrieval (w.o. $L_\text{dm}$) | IMSE-Retrieval |
> | --- | --- | --- | --- | --- | --- | --- |
> | Time / Batch(Sec) | 0.14 | **0.31** | 3.49 | 2.52 | 0.84 | 0.99 |
> | Accuracy | 51.0 | 52.8 | 57.7 | 50.7 | 62.8 | **64.4** |

---

### Official Review · Reviewer_4qVn · 2025-11-01

**Soundness:** 3
**Presentation:** 3
**Contribution:** 3
**Rating:** 6
**Confidence:** 4

**Summary:**

This paper proposes Intrinsic Mixture of Spectral Experts, a novel framework for test-time adaptation. IMSE updates only the singular values while keeping the singular vectors fixed, thus enabling parameter-efficient adaptation. To prevent the entropy minimization objective from collapsing feature diversity, the authors introduce a diversity maximization loss based on spectral vector–input alignment. For CTTA, they propose a Domain-Aware Spectral Code Retrieval mechanism, which stores domain-specific singular values and retrieves them according to domain similarity to mitigate forgetting. Experiments on ImageNet-C  and CLIP  backbones show that IMSE achieves state-of-the-art results while using fewer trainable parameters.

**Strengths:**

1. The paper correctly identifies two practical problems in TTA: the overfitting tendency of entropy minimization and catastrophic forgetting in CTTA.
2. The idea of reinterpreting linear layers as mixtures of rank-1 “spectral experts” is interesting. It provides a compact and interpretable way to perform fine-grained adaptation.
3. Updating only singular values makes IMSE computationally efficient and easily applicable to existing pretrained models such as ViT or CLIP.
4. The reported improvements on ImageNet-C and the low parameter count are convincing, and the experiments are well organized.

**Weaknesses:**

1. How robust is the retrieval process when encountering a completely new domain that differs substantially from all stored domains?
2. Updating only singular values implicitly constrains parameter updates to a diagonal submanifold of the low-rank space. Does this constraint reduce optimization expressivity when confronted with severe domain shifts?
3. How much extra computational cost is introduced by SVD?

**Questions:**

See above.

---

> ### Author Response · Authors · 2025-11-21
>
> We thank the reviewer for the positive assessment of our work and for highlighting the strengths of the proposed IMSE framework. We carefully address the reviewer’s concerns regarding retrieval robustness under unseen domains, the expressivity of singular-value only adaptation under severe domain shifts, and the computational overhead introduced by SVD. We conduct additional experiments and provide detailed analyses to clarify these points.
>
> ---
>
> ### **W1 - Robustness of the Retrieval Process**
>
> When a completely new domain appears during continual test time adaptation, the proposed method detects a domain change based on its discrepancy from the previous domain. The newly observed domain descriptor is then compared with all descriptors in the domain bank, and the most similar one is selected for retrieval. For such unseen domains, the similarity scores to all stored descriptors are low, so the closest domain is retrieved only as an initialization.
>
> The subsequent adaptation steps then adjust the singular values to rapidly align the model with the new distribution. For example, as shown in Table 4, when the sequence shifts from Gaussian, shot, and impulse noise to a different corruption type such as Defocus Blur, IMSE-Retrieval selects the pretrained domain, allowing the model to adapt rapidly and stably to the new domain. These results demonstrate that the **retrieval process remains effective even when the incoming domain differs substantially from all stored domains**.
>
> ---
>
> ### **W2 : More Severe Domain Shifts**
>
> Our method updates only the singular values while keeping the direction of spectral experts fixed, enabling parameter-efficient adaptation. To demonstrate the effectiveness of this approach under more severe domain shifts, we evaluate IMSE on benchmarks beyond ImageNet-C, including ImageNet-V2 (distribution shifts), Sketch (texture shifts), ImageNet-3DCC-severity 5 (geometry-aware real world corruptions), and OfficeHome & DomainNet (large cross-domain gaps). Specifically, ImageNet-3DCC [1] is more challenging than standard 2D common corruptions [2]. Its corruptions are generated using 3D scene geometry, resulting in more realistic and real-world–plausible distribution shifts. As shown in Tables 2-A, 2-B, 2-C, and 2-D, IMSE achieves state-of-the-art performance on average across all benchmarks. These results suggest that **adapting only the singular values is acceptable to capture even large and challenging domain shifts**, without the need to update the basis directions.
>
> Table 2-A. Test-time adaptation on ImageNet-V2/Sketch
>
> ||ImageNet-V2|ImageNet-Sketch|
> |---|---|---|
> |Source|73.83|43|
> |TENT|74.28|50.13|
> |SAR|74.35|52.1|
> |DPAL|74.4|52.25|
> |IMSE|**74.48**|**53.58**|
>
> Table 2-B. Test-time adaptation on ImageNet-3DCC, accuracy across 11 corruption domains and the average performance.
>
> | Method | bit error | color quantization | far focus | flash | fog | h265 abr | h265 crf | iso noise | low light | near focus | xy motion blur | z motion blur | Avg. |
> | --- | --- | --- | --- | --- | --- | --- | --- | --- | --- | --- | --- | --- | --- |
> |Source|**20.3**|53.1|61.7|47.3|39.4|52.8|57.3|59.1|56.3|70.5|43.2|47.4|50.7|
> |TENT|1.2|64.1|68.2|52.1|7.5|56.1|60.7|65.6|70.3|76.2|56.4|61.2|53.3|
> |SAR|13.8|64.7|68.7|54.3|41.4|57.1|61.4|66.3|70.8|76.4|57.8|62.5|57.9|
> |DPAL|13.1|64.2|68.3|53.7|43.4|57.0|61.2|65.8|70.0|75.8|56.6|61.7|57.6|
> |IMSE|15.5|**65.0**|**70.1**|**55.2**|**44.5**|**58.1**|**62.2**|**67.3**|**71.6**|**77.1**|**58.9**|**63.9**|**59.1**|
>
> Table 2-C. Test-time adaptation on OfficeHome, accuracy across 12 source and target domain pairs and average performance.
>
> ||R->A|R-C|R->P|A->R|A->C|A->P|C->R|C->A|C->P|P->R|P->A|P->C|Avg.|
> |---|---|---|---|---|---|---|---|---|---|---|---|---|---|
> |Source|79.68|56.05|88.35|87.46|58.55|82.9|84.66|78.37|84.29|87.74|75.32|55.69|76.58|
> |TENT|79.68|56.08|**88.91**|88.01|59.61|83.1|85.42|**79.68**|**85.01**|87.9|75.48|55.71|77.05|
> |SAR|79.72|**56.72**|88.37|87.62|**61.42**|83.03|85.4|79.48|84.61|87.81|76.02|56.26|77.20|
> |DPAL|79.93|55.57|86.52|87.14|59.17|82.61|85.33|79.02|84.09|86.68|75.85|55.25|76.43|
> |IMSE|**81.04**|56.35|**88.91**|**88.13**|60.36|**83.44**|**85.74**|79.39|84.88|**88.04**|**76.14**|**56.28**|**77.39**|
>
> Table 2-D. Test-time adaptation on DomainNet, accuracy across 12 source and target domain pairs and average performance.
>
> ||R->C|R->P|R->S|C->R|C->P|C->S|P->R|P->C|P->S|S->R|S->C|S->P|Avg.|
> |---|---|---|---|---|---|---|---|---|---|---|---|---|---|
> |Source|66.84|76.14|59.44|80.46|69.85|65.93|86.72|69.04|**59.95**|82.19|72.95|**73.62**|71.93|
> |TENT|68.25|76.45|60.36|81.91|72.32|67.01|87.6|69.97|59.21|82.76|73.05|73.11|72.67|
> |SAR|**69.64**|**77.78**|**64.23**|81.91|72.83|**67.73**|86.72|**70.77**|59.41|82.77|73.53|73.41|73.39|
> |DPAL|68.14|77.54|60.86|81.78|72.79|66.07|84.93|70.16|58.86|81.61|72.54|70.11|72.12|
> |IMSE|69.33|**77.78**|62.4|**82.45**|**73.23**|67.57|**87.85**|70.42|59.72|**83.1**|**73.73**|73.55|**73.43**|

---

> ### Author Response · Authors · 2025-11-21
>
> ### **W3 : Extra Computational Cost of SVD**
>
> The singular value decomposition (SVD) of all linear layers is performed only once as an offline preprocessing step before adaptation begins. Quantitatively, performing **full SVD on all linear layers of ViT-Base takes approximately 5.0 seconds on a single A6000 GPU**, which is negligible compared to the duration of the continual test-time adaptation process. Second, during the adaptation phase, the weight reconstruction ($W = U\Sigma V^T$) does introduce additional matrix multiplications. However, this overhead of 0.07 seconds per batch is marginal compared to the 0.99 seconds required for the full adaptation pass, accounting for only about **7% of the total adaptation time**.
>
> ---
>
> ### **Reference**
>
> [1] Kar, Oğuzhan Fatih, et al. "3d common corruptions and data augmentation." *Proceedings of the IEEE/CVF Conference on Computer Vision and Pattern Recognition*. 2022.
>
> [2] Hendrycks, Dan, and Thomas Dietterich. "Benchmarking Neural Network Robustness to Common Corruptions and Perturbations." *International Conference on Learning Representations*.

---

### Official Review · Reviewer_23Fr · 2025-11-03

**Soundness:** 4
**Presentation:** 3
**Contribution:** 3
**Rating:** 8
**Confidence:** 3

**Summary:**

The paper introduces IMSE (Intrinsic Mixture of Spectral Experts), a new framework for continual test-time adaptation (CTTA). IMSE decomposes each linear layer in Vision Transformers using singular value decomposition (SVD), interpreting the orthogonal basis matrices as a mixture of spectral experts and the diagonal matrix of singular values as spectral weights. During adaptation, IMSE fine-tunes only these spectral weights while keeping the orthogonal bases fixed, enabling highly parameter-efficient adaptation.

To guide unsupervised adaptation, the authors propose a combined loss function that integrates entropy minimization with a diversity maximization term to prevent feature collapse. IMSE achieves state-of-the-art performance on ImageNet-C, -R, and -A benchmarks, outperforming strong TTA baselines such as TENT, SAR, and DPAL while requiring significantly fewer trainable parameters.

For continual TTA, the paper further introduces a Domain Bank that stores domain-specific spectral codes and descriptors, enabling retrieval and reuse of prior adaptations to mitigate catastrophic forgetting. Comprehensive ablation studies demonstrate the contribution of each proposed component.

**Strengths:**

Originality: IMSE offers a novel interpretation of test-time adaptation through the lens of spectral experts, extending SVD-based parameter-efficient fine-tuning (e.g., LoRA, SVDiff) to the unsupervised TTA setting. The idea of freezing orthogonal bases and adapting only singular values is conceptually simple yet highly effective.

Efficiency and Practicality: By updating only singular values, IMSE achieves orders-of-magnitude parameter reduction (~2000× fewer trainable parameters) while maintaining or exceeding SOTA performance.

Continual Adaptation Innovation: The introduction of the Domain Bank provides a simple but powerful mechanism for mitigating catastrophic forgetting across sequential domains.

Comprehensive Ablation Analysis: The paper includes well-designed ablation studies that isolate the impact of each component.

**Weaknesses:**

Domain Bank scalability:
Using simple KL divergence over mean–variance descriptors might struggle to discriminate fine-grained domain shifts, potentially leading to incorrect retrievals.

No direct measure of forgetting:
Although the Domain Bank is designed to mitigate catastrophic forgetting, the paper does not include quantitative evidence, e.g. Backward Transfer (BWT), to verify that performance on previously adapted domains remains stable after subsequent adaptations.

**Questions:**

Could the authors quantify the computational cost of performing full SVDs across all linear layers in large-scale ViTs (e.g., CLIP or MAE)? Have they considered using truncated or randomized SVD to reduce preprocessing overhead, and if so, how would this affect adaptation performance?

---

> ### Author Response · Authors · 2025-11-21
>
> We sincerely thank the reviewer for the thoughtful and constructive feedback, as well as the positive assessment of our work. We appreciate the reviewer’s recognition of the strengths of IMSE and the detailed comments that helped us clarify important aspects of the method. In the following, we address each concern point-by-point and provide additional experiments and explanations to further strengthen the paper.
>
> ---
>
> ### **W1 : Domain Bank Scalability**
>
> To evaluate whether the proposed method can reliably discriminate fine-grained domain shifts, we conduct an additional gradual shift experiment. Following the same domain order of Table 4 in the manuscript (e.g., Gaussian Noise (GN) → Shot Noise → Impulse Noise (IN) → Defocus Blur (DB) → … → JPEG Compression), we construct a sequence where the severity level gradually changes as 1→2→3→4→5→4→3→2→1 within each corruption type. With 15 corruption types repeated 9 times, this setting produces 135 sequential test domains.
>
> As shown in Table 1-A, **IMSE-Retrieval achieves the highest performance.**
>
> Table 1-A. Continual test-time adaptation on ImageNet-C under gradual shifts.
>
> | Method | Average accuracy. |
> | --- | --- |
> | Source | 67.3 |
> | TENT | 70.7 |
> | CoTTA | 69.5 |
> | ViDA | 72.5 |
> | IMSE-Retrieval | **74.9** |
>
> We additionally analyze its retrieval behavior using a domain change threshold of $\tau=0.2$, as detailed below.
>
> 1. **GN1→…→GN5 (increasing severity):** IMSE-Retrieval consistently retrieves the spectral code corresponding to the closest severity level.
> 2. **GN5→…→GN1 (decreasing severity):** IMSE-Retrieval retrieves the encountered domains in the increasing stage.
> 3. **Transition from IN1 to DB1:** IMSE-Retrieval correctly detects a domain change and selects the pretrained domain as the most similar domain.
> 4. **DB1→DB5→DB1:** Retrieval process remains stable and consistent, as observed in the preceding scenario
>
> These results suggest that the descriptor remains responsive under gradual distribution changes.
>
> Furthermore, to address the over-segmentation issue, we increase the domain-change threshold $\tau$ and observe how this affects both domain-change detection quality and adaptation performance. In evaluating domain-change detection, we allow a margin of one step around each transition point to account for the fine-grained nature of gradual shifts. When the threshold is set to a more tolerant value ($\tau = 0.04$), the F1 score of domain-change detection decreases F1: 0.77 **→** 0.28), while the overall performance increases (accuracy: 74.9 **→** 75.6). This trade-off indicates that sensitive domain change detection is not always advantageous, and that choosing a moderate threshold helps achieve strong performance while keeping the Domain Bank compact.
>
> ---
>
> ### **W2 : Measure of Forgetting**
>
> Although measuring Backward Transfer (BWT) is important for evaluating forgetting in standard continual learning (CL), the metric is not a primary consideration in the Continual test-time adaptation (CTTA) setting.  BWT assumes a sequential training process in which the model’s parameters evolve continuously across tasks, and forgetting is quantified by how much later updates degrade earlier tasks. In contrast, our method does not update parameters in a strictly sequential manner. Each time a domain arrives, we identify its the most similar domain and initialize the model using the corresponding spectral code stored in the Domain Bank. As a result, previously stored codes are not overwritten by adapting on unrelated domains. Because of this design, BWT does not meaningfully capture forgetting in our framework.
>
> Instead, evaluating the model in a recurring-domain scenario, where previously seen domains reappear, could serve as an alternative metric. To evaluate this, we conduct an additional recurring-domain experiment in which the full corruption sequence of Table 4 (Gaussian noise → … → JPEG compression) is repeated. As shown in Table 1-B, **IMSE-Retrieval consistently achieves the highest accuracy** on recurring domains. This demonstrates that the IMSE-Retrieval mitigates forgetting and enables fast and stable re-adaptation when previously encountered domains reappear.
>
> Table 1-B Continual test-time adaptation on Imagenet-C, average accuracy across 2 rounds.
>
> | Method | Round 1 | Round 2 |
> | --- | --- | --- |
> | Source | 50.1 | 50.1 |
> | TENT | 52.8 | 55.3 |
> | CoTTA | 50.6 | 49.7 |
> | ViDA | 57.7 | 59.5 |
> | IMSE-Retrieval | **64.4** | **65.7** |

---

> ### Author Response · Authors · 2025-11-21
>
> ### **Q1 : Computational Cost of SVD**
>
> All singular value decompositions across the linear layers are performed once as an offline preprocessing step before adaptation. For ViT-Base, this **full SVD computation takes approximately 5.0 seconds on a single A6000 GPU**, which represents a very small portion of the overall computational cost in CTTA. We also consider truncated SVD for reducing the the preprocessing cost. In particular, truncated SVD is evaluated as a strategy for reducing the number of trainable parameters, as analyzed in Section 5 (Effect of Singular Value Selection Strategy). As shown in Figure 2(a), adapting only the top 50 percent of singular values yields almost identical adaptation performance to using the full spectrum. These results show that the cost of full SVD computation is already minimal and that truncated SVD provides an additional efficiency benefit without noticeable performance degradation.
>
> Additionally, during the adaptation phase, the weight reconstruction ($W = U\Sigma V^T$) does introduce additional matrix multiplications.  However, this overhead of 0.07 seconds per batch is marginal compared to the 0.99 seconds required for the full adaptation pass, accounting for only about **7% of the total adaptation time**. Therefore, both the one-time SVD preprocessing and the per-step reconstruction cost remain negligible relative to the overall CTTA computation.

---

### Author Response · Authors · 2025-12-01
**Summary of Key Contributions, Response, and Discussion (1/2)**

Dear Area Chair and Reviewers,

We deeply appreciate the reviewers' dedicated time and constructive feedback, which is invaluable in refining our manuscript. We also extend our deepest gratitude to the Area Chair for handling this submission under these unprecedented and challenging circumstances.

To provide the AC with a summary of the prior discussions and the key contributions of our work, we organize this final response into the following three sections:

- **Key Contributions:** Summarizing the novelty and validity of the proposed method.
- **Response Summary:** Detailing how we addressed key concerns and manuscript updates.
- **Discussion Summary:** Highlighting the positive consensus formed during the discussion, including a **confirmed score increase (Reviewer Q6jF)** and **fulfilled conditions for re-evaluation (Reviewer wHid)**.

### **1. Key Contributions**

The key contributions of our work, acknowledged as strengths by the reviewers, are summarized as follows:

- **Leveraging of Pre-trained Feature Extractors:** We propose a novel framework utilizing intrinsic spectral experts to fully exploit pre-trained feature extractors **(Originality: Reviewer 23Fr,  Strength: Reviewer 4qVn, Q6jF)**.
- **Alleviation of Feature-Diversity Collapse:** We address feature-diversity collapse during TTA by diversifying the response patterns of specific spectral experts **(Strength: Reviewer 4qVn, wHid)**.
- **Explicit Mitigation of Forgetting:** We introduce Domain-aware Spectral code Retrieval via a compact Domain Bank, which explicitly mitigates catastrophic forgetting during CTTA **(Innovation: Reviewer 23Fr, Strength: Reviewer 4qVn, wHid)**.
- **Comprehensive Validation & Efficiency:** Our method demonstrates superior performance and efficiency across diverse pre-trained models, challenging scenarios, and standard DA benchmarks **(Efficiency: Reviewer 23Fr, Strength: Reviewer 4qVn)**.

### **2. Response Summary**

Reviewers raised constructive questions regarding performance under fine-grained [Reviewer 23Fr-W1, Reviewer Q6jF-Q3 & Q6] or severe domain shifts [Reviewer 4qVn-W2, Reviewer Q6jF-W1 & Q1], long-term adaptation [Reviewer Q6jF-W3], and standard DA benchmarks [Reviewer wHid-W2]. Other key inquiries addressed the efficiency of the proposed method [Reviewer 23Fr-Q1, Reviewer 4qVn-W3, Reviewer wHid-W4], the impact of specific components via ablation studies [Reviewer wHid-W3, Reviewer Q6jF-W2 and Q2], and concerns regarding the diversity maximization [Reviewer Q6jF-Q4].

To address these concerns, we conducted extensive new experiments and revised the manuscript accordingly. Key updates are as follows:

- **Broader Applicability (Gradual & Severe Shifts):** We verified our method in the fine-grained (gradual) shift scenario (**Section 4.5**) and evaluated IMSE on diverse and challenging benchmarks including ImageNet-3DCC, ImageNet-V2/Sketch, OfficeHome, and DomainNet **(Section I in supp)**.
- **Long-term Stability:** We conducted additional experiments to confirm the robustness of our method in the long-term continual adaptation scenario **(Section J in supp)**.
- **Efficiency Analysis:** We provided a detailed analysis of runtime, additional storage, and SVD overhead to demonstrate the practicality of IMSE-Retrieval (**Section 5**).
- **Component Ablation Studies:** We validated the contribution of each individual component through comprehensive ablation studies (**Section G in supp**).
- **Robustness of Diversity Maximization:** To address concerns about task-feature suppression, we verified the effectiveness of our diversity maximization even in non-i.i.d. settings **(Section H in supp).**

---

> ### Author Response · Authors · 2025-12-01
> **Summary of Key Contributions, Response, and Discussion (2/2)**
>
> ### **3. Discussion Summary**
>
> We appreciate the reviewers' positive response regarding the novelty and contribution of our proposed method.
>
> The constructive discussions allowed us to further strengthen the validity of our work, leading to a more favorable assessment.
>
> In the discussion period, we addressed concerns of Reviewer Q6jF regarding challenging domain-shift scenarios, scalability, component contributions, and diversity maximization by providing comprehensive validation on challenging benchmarks (ImageNet-3DCC, OfficeHome, DomainNet, etc.), diverse scenarios (including gradual and long-term CTTA), component ablation studies, and evaluation under non-i.i.d. settings. **Consequently, Reviewer Q6jF stated in their comment that their earlier concerns had been addressed, and accordingly raised their score to 8 (Accept) on Nov 23 at 3:48 PM UTC.**
>
> Furthermore, we have provided all experiments requested by Reviewer wHid, including component ablation studies, DA benchmark results, and efficiency analysis. These results demonstrate the superiority of our method, confirming that it outperforms CTTA baselines in both performance and efficiency. Although **Reviewer wHid** has not yet participated in the discussion, given the reviewer’s explicit statement in the initial review (”***If you can conduct additional experiments to further evaluate your method, I would be willing to raise my score”)***, **we believe that our response fully satisfies the conditions required for a positive re-evaluation.**
>
> While the remaining two reviewers (**Reviewer 23Fr, Reviewer 4qVn**) have not yet participated in the discussion, we have provided comprehensive responses including extensive additional experiments and detailed efficiency analysis. We are confident that these materials effectively address their concerns. Therefore, **we believe that these reviewers would likely maintain their positive assessments or raise their scores based on our comprehensive response.**
>
> ### **Conclusion**
>
> We are confident that the additional experiments, analysis, and clarifications provided during the discussion period **effectively address the key concerns** raised by the reviewers, and these updates have been **fully incorporated into the revised manuscript**. We believe that our work presents a **novel and highly efficient CTTA framework supported by extensive empirical validation**, and we respectfully look forward to your positive assessment.
>
> Sincerely,
>
> The Authors

---

### Meta-Review · Area_Chair_eTE9 · 2025-12-28

**Summary:**

The paper proposes a parameter-efficient TTA/CTTA framework that decomposes pretrained linear layers via SVD and adapts only the singular values, combined with diversity regularization and a domain-retrieval mechanism for continual adaptation. Reviewers generally viewed the spectral-expert formulation as novel and practically relevant, and the empirical results demonstrate strong performance with favorable efficiency. The rebuttal and discussion substantially strengthened the validation and addressed the main concerns raised in the reviews.
The initial reviewer scores were 8, 6, 4, and 4, and one reviewer with an initial score of 4 explicitly indicated willingness to raise their initial score after the rebuttal. Based on these observations, I recommend Accept.

**Reviewer Concerns:**

The rebuttal addressed the main concerns raised by multiple reviewers about whether updating only singular values is expressive enough and whether this design limits adaptation under severe domain shifts. These concerns were resolved by adding experiments on more challenging benchmarks and by clearly explaining the role of fixed spectral bases and how each spectral expert responds to input data.
Questions about the robustness and scalability of the domain bank were also addressed through additional experiments under gradual. domain shifts and long-term or recurring CTTA settings, showing stable adaptation and reduced forgetting. Concerns about the role and. necessity of each loss component were clarified with new ablation studies that separately analyzed entropy minimization, diversity. maximization, and the retrieval mechanism.
Finally, efficiency-related concerns, including the cost of SVD, runtime, and memory usage, were addressed with concrete measurements showing that the overhead is small and practical.

**Reviewer Scores:**

This submission received initial scores of 8, 6, 4, and 4. Given that most key concerns were resolved during the rebuttal and discussion, and that one reviewer with an initial score of 4 explicitly indicated willingness to raise their score, I recommend Accept.

---

### Decision · Program_Chairs · 2026-01-26

Accept (Poster)